# The Cep57-pericentrin module organizes PCM expansion and centriole engagement

Koki Watanabe[1,2,3], Daisuke Takao[1,3], Kei K Ito[3], Mikiko Takahashi[4] & Daiju Kitagawa[1,2,3]

Centriole duplication occurs once per cell cycle to ensure robust formation of bipolar spindles and chromosome segregation. Each newly-formed daughter centriole remains connected to its mother centriole until late mitosis. The disengagement of the centriole pair is required for centriole duplication. However, the mechanisms underlying centriole engagement remain poorly understood. Here, we show that Cep57 is required for pericentriolar material (PCM) organization that regulates centriole engagement. Depletion of Cep57 causes PCM disorganization and precocious centriole disengagement during mitosis. The disengaged daughter centrioles acquire ectopic microtubule-organizing-center activity, which results in chromosome mis-segregation. Similar defects are observed in mosaic variegated aneuploidy syndrome patient cells with *cep57* mutations. We also find that Cep57 binds to the well-conserved PACT domain of pericentrin. Microcephaly osteodysplastic primordial dwarfism disease *pericentrin* mutations impair the Cep57-pericentrin interaction and lead to PCM disorganization. Together, our work demonstrates that Cep57 provides a critical interface between the centriole core and PCM.

[1] Division of Centrosome Biology, National Institute of Genetics, Mishima, Shizuoka 411-8540, Japan. [2] Department of Genetics, School of Life Science, The Graduate University for Advanced Studies (SOKENDAI), Hayama, Kanagawa 240-0193, Japan. [3] Department of Physiological Chemistry, Graduate school of Pharmaceutical Sciences, The University of Tokyo, Bunkyo, Tokyo 113-0033, Japan. [4] Faculty of Pharmaceutical Sciences, Teikyo Heisei University, Nakano, Tokyo 164-8530, Japan. Correspondence and requests for materials should be addressed to D.K. (email: dkitagawa@mol.f.u-tokyo.ac.jp)

Centrosomes are non-membrane-bound organelles that serve as the major microtubule-organizing centers (MTOCs) in most animal cells and participate in diverse biological processes such as cell division and motility[1–4]. A single centrosome consists of two centrioles and a surrounding amorphous protein matrix known as pericentriolar material (PCM). Abnormalities in centrosome organization and function often result in genomic instability and aberrant cell division. Indeed, mutations in many centrosomal proteins have also been implicated as a cause of cancer and autosomal recessive disorders[5,6].

The centriole duplication cycle is tightly regulated and coupled with cell cycle progression[1,7]. Toward the G1-to-S transition, centriole formation begins with the assembly of the cartwheel structure that mainly dictates the universal radial nine-fold symmetry of centrioles, followed by attachment of peripheral centriolar microtubules[8]. Toward the end of G2, the proteinaceous linker connecting the two mother centrioles is dissolved and the two centrosomes migrate to the opposite ends of the cell. During mitosis, the centrosomes act as MTOCs to ensure the robust formation of mitotic bipolar spindle and proper chromosome segregation. At this stage, surrounding PCM drastically expands and acquires MTOC activity. Each newly formed daughter centriole is orthogonally connected to each mother centriole until late mitosis (centriole engagement). The loss of connection between the mother and daughter centrioles occurs after cytokinesis with the disassembly of expanded PCM. The disconnection process is called centriole disengagement and thought to be a licensing step for centriole duplication in the next cell cycle[9–11]. Therefore, the timing of centriole disengagement must be tightly regulated. However, the mechanisms underlying centriole engagement remain elusive.

Recently, it has been suggested that PCM integrity is prerequisite for centriole engagement[12–14]. However, how surrounding PCM contributes to centriole engagement and conversely how centrioles facilitate formation of the highly organized PCM structure are still poorly understood. The centriole disengagement that normally occurs towards the mitotic exit, requires the activity of Plk1, a mitotic kinase, and separase in vertebrates[11,15,16]. Pericentrin (PCNT, also known as kendrin), a PCM component, is known to be a critical substrate cleaved by separase for centriole disengagement[9,10]. This cleavage event in mitosis is necessary for timely centriole disengagement and for licensing a new round of centriole duplication in the next cell cycle. Furthermore, phosphorylation of PCNT by Plk1 seems to be a priming step for separase-dependent cleavage of PCNT in mitosis[17]. However, given that PCNT is also involved in expansion of mitotic PCM[18], how PCNT regulates these two crucial events in human centrosome biogenesis remains unclear.

Previous studies reported that centrosomal protein of 57 kDa (Cep57) is responsible for mosaic variegated aneuploidy (MVA) syndrome and is supposed to be required for proper chromosome segregation[19,20]. It has been recently reported that Cep57 regulates the loading of spindle assembly checkpoint proteins, the Mad1–Mad2 complex, at kinetochores for timely chromosome segregation in human cells[21]. However, it is controversial whether Cep57 is a kinetochore component or a centrosomal protein[22,23]. Indeed, Cep57 is also known to be a PCM component that is critical for the proper organization of spindle microtubules and recruitment of spindle focusing proteins[24]. The previous work indicated that Cep57 depletion resulted in multipolar spindle formation presumably due to PCM fragmentation[24]. Chemical crosslinking experiments revealed that Cep57 forms a complex with Cep152 and Cep63 around the proximal end of centrioles in human interphase cells[25]. However, the exact function of Cep57 in human centrosome biogenesis remains to

be elucidated. Moreover, it is not clear whether its functional homologs in other species also function in a similar fashion.

In this study, we reveal that Cep57, an evolutionarily conserved protein, is required for PCM organization that regulates centriole engagement. Depletion of Cep57 causes PCM disorganization and precocious centriole disengagement during mitosis. Intriguingly, the disengaged daughter centrioles acquire ectopic MTOC activity, which results in chromosome mis-segregation and aneuploidy. MVA patients' cells carrying Cep57 mutations also exhibit similar defects, such as precocious centriole disengagement, suggesting a potential cause of the MVA disease. We also find that Cep57 localizes at mother centrioles and binds to the well-conserved PACT domain of pericentrin (PCNT). Importantly, microcephaly osteodysplastic primordial dwarfism disease-related PCNT mutations impair the Cep57–PCNT interaction and lead to PCM disorganization. Overall, these findings lead us to propose that the Cep57–PCNT complex provides a critical interface between the centriole core and PCM, which is crucial for the higher-order structure of the PCM that holds a mother–daughter pair of centrioles.

## Results

**Cep57 is an evolutionarily conserved component and forms ring-like structures around the mother centriole wall**. To determine whether Cep57 is a conserved factor across species, we first performed BLAST analysis. We succeeded in identifying a homologous 36 amino-acid (a.a.) sequence in the N-terminal coiled-coil region of Cep57 family proteins in vertebrates. The conserved sequence was also found in T. cruzi. Unfortunately, our BLAST search did not hit any other invertebrate organisms presumably due to the short consensus sequence. Accordingly, we designated the conserved sequence the "PINC (present in N-terminus of Cep57)" motif (Fig. 1a, b, Supplementary Fig. 1a).

Next, to investigate the precise distribution of Cep57 at centrosomes across the cell cycle, we performed an immuno-fluorescence analysis with HeLa cells by using stimulated emission depletion (STED) microscopy. STED analysis revealed that Cep57 formed ring-like structures around the proximal end of the mother centriole wall across the cell cycle (Fig. 1c). We then sought to compare the diameter of centriolar rings of Cep57 with that of Cep192 rings that mark PCM. The diameter of the Cep57 ring was close to that of Cep192 (Cep57, 219.9 ± 13.9 nm; Cep192, 221.8 ± 18.5 nm; values are mean ± s.d., $n = 10$), suggesting that Cep57 is a component of the inner layer of the PCM (Fig. 1d, e). We also found that Cep57 was newly recruited to new mother centrioles gradually during the G1 phase, whereas Cep57 was confined to old mother centrioles just after cytokinesis and in early G1 (Fig. 1f, Supplementary Fig. 1b). We noted that the difference between Cep57 signal intensities at old and new mother centrioles during the G1 phase was due to the difference in the height of Cep57 that bound along the side of the centriole wall (Fig. 1f, Supplementary Fig. 1c). Furthermore, the signal intensity of Cep57 at new mother centrioles is proportional to that of PCNT, a major component of PCM ($R^2 = 0.70$), suggesting that both Cep57 and PCM components were simultaneously recruited around new mother centrioles in the G1 phase (Fig. 1g, h). We next aimed to identify which domains of Cep57 are required for its centriolar localization. Consistent with previous reports[23], the N-terminal half of the Cep57 fragment containing the first coiled-coil domain localized to centrioles. We further narrowed down the region required for centriolar localization of Cep57 in the first coiled-coil domain. Since secondary structure prediction analysis identified three discrete coiled-coil motifs in the first coiled-coil-enriched domain (Supplementary Fig. 1d), we then constructed the deletion

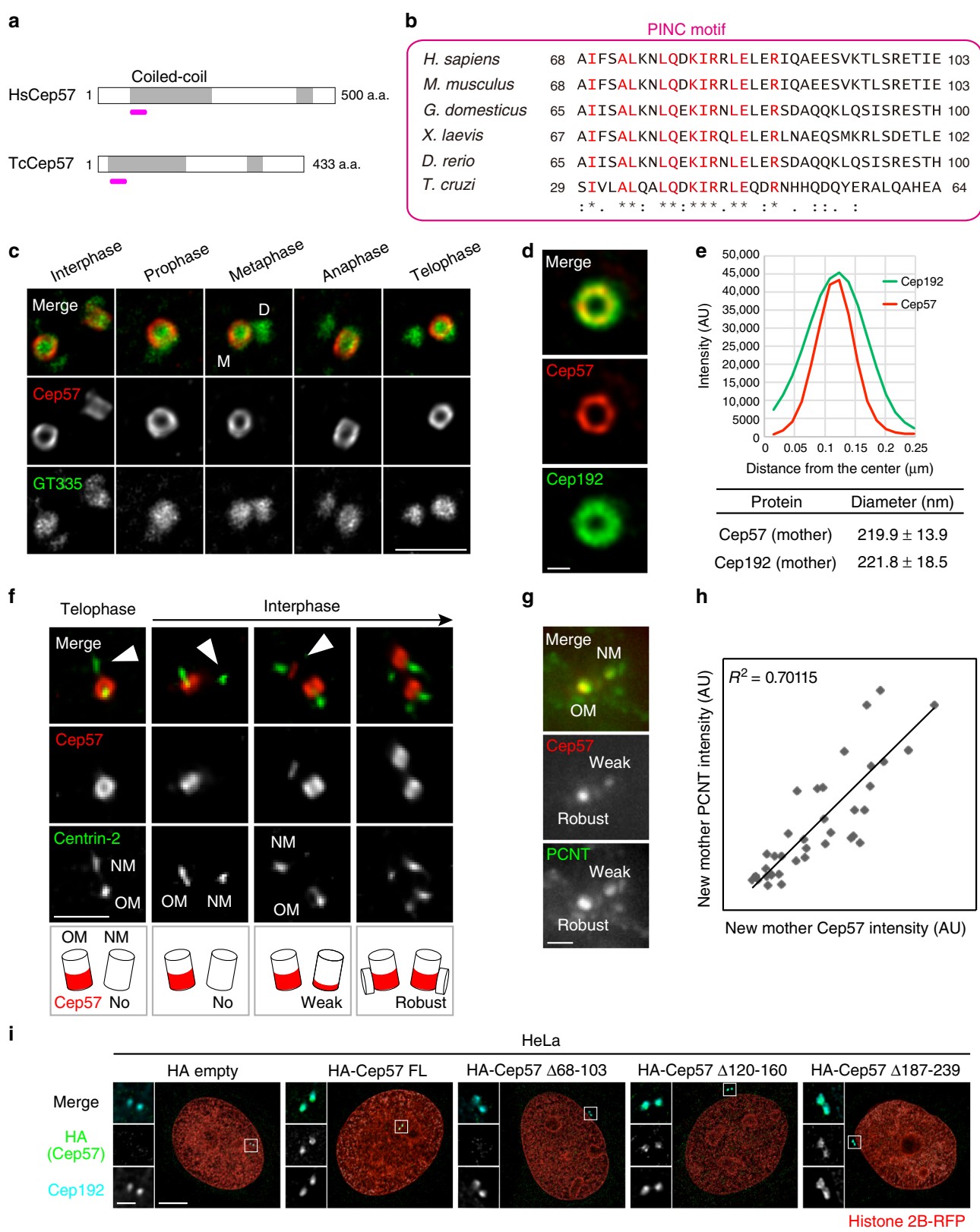

mutants that lack each coiled-coil motif. Whereas the two Cep57Δ120–160 and Cep57Δ187–239 deletion mutants localized to centrioles, a Cep57 deletion mutant lacking the PINC domain (Cep57Δ68–103) failed to localize to centrioles (Fig. 1i). We also noted that overexpressed Cep57 occasionally localized to the filamentous microtubule network around the nucleus, which is consistent with previous reports[23,24] (Supplementary Fig. 1e). Overall, these findings suggest that Cep57 is an evolutionarily conserved factor across species and localizes around the proximal part of the mother centriole wall.

**Fig. 1** Cep57 is an evolutionarily conserved component and forms ring-like structures around the mother centriole wall. **a** Schematic diagrams of *H. sapiens* Cep57 (HsCep57) and *T. cruzi* Cep57 (TcCep57). Coiled-coil domains are shown in gray box. The position of evolutionarily conserved domain (PINC motif) is indicated in pink line. **b** Alignments of the evolutionarily conserved domain (PINC motif) within *H. sapiens*, *M. musculus*, *G. domesticus*, *X. laevis*, *D. rerio*, and *T. cruzi*. Identical residues determined by Clustal Omega are shown in red. Asterisks indicate the residues identical in all aligned sequences; colons: conserved substitutions; periods: semi-conserved substitutions. **c** Centriolar distribution of Cep57 at different cell cycle stages. HeLa cells were immunostained with antibodies against Cep57 (red) and GT335 (green) and observed by STED microscopy. Scale bar, 1 μm. **d** STED images representing top views of Cep57 and Cep192 at mother centrioles. Scale bar, 200 nm. **e** The graph shows radial profiles from the center of the Cep57 and Cep192 rings. The obtained profile was then fitted with a Gaussian curve and the distance between the center of the ring and the peak of the Gaussian curve was defined as the radius. The diameter of the ring was defined as twice the radius. Values are mean distance ± s.d. ($n = 10$). **f** Cep57 becomes gradually enriched at new mother centrioles in interphase. HeLa cells were immunostained with antibodies against Cep57 and centrin-2. Arrowheads indicate new mother centrioles (OM old mother, NM new mother). Scale bar, 1 μm. **g** The signal intensity of Cep57 at new mother centrioles is proportional to that of PCNT. HeLa cells were immunostained with antibodies against Cep57 and PCNT. Scale bar, 1 μm. **h** Dot plots represent quantification of the signal intensity of Cep57 and PCNT at new mother centrioles ($n = 36$). **i** HeLa cells co-expressing HA empty (control), HA-Cep57, or the indicated HA-Cep57 deletion mutants (1 μg) and Histone 2B-RFP (transfection control, 10 ng) were immunostained with antibodies against HA (green), Cep192 (cyan) and RFP (red). Scale bar, 5 μm in the low-magnified view, 1 μm in the inset

**Cep57 is essential for PCM organization and centriole disengagement in mitosis**. To investigate the function of Cep57, we next depleted endogenous Cep57 using short interfering RNAs (siRNAs) in HeLa cells, and found that depletion of Cep57 caused PCM disorganization (74.7 ± 3.8%, from three experiments) and precocious centriole disengagement (68.7 ± 2.5%, from three experiments) in mitotic cells (Fig. 2a–c). We discriminated precocious centriole disengagement in Cep57-depleted cells when the distance between mother and daughter centrioles, in prometaphase or metaphase, was at least twice more than that of the control or interphase cells. In contrast, depletion of Cep57 did not lead to any defects in centriolar recruitment of Cep152, centriole duplication, and PCM organization in interphase cells (Supplementary Fig. 2a–c). To ensure the two centrioles seen at the poles are the mother and daughter centriole pair, we tested ODF2 staining which marks mother centrioles and found a single ODF2-positive focus out of two separate centrioles at a spindle pole in Cep57-depleted cells (Supplementary Fig. 2d). This observation indicates that the two centrioles seen at the pole are a pair of centrioles. The same mitotic defects were observed in different human cell lines (Supplementary Fig. 2e) or when using siRNAs targeting different sequences of Cep57 ORF (Supplementary Fig. 2f). The efficacy of Cep57 siRNA was confirmed using immunofluorescence analysis with specific antibodies against endogenous Cep57 (Supplementary Fig. 2g). Furthermore, these defects were rescued by expressing RNAi-resistant full-length Cep57 constructs, indicating that the phenotypes observed were not due to the off-target effects of Cep57 siRNA (Supplementary Fig. 3a, b). To further confirm the phenotypes upon depletion of Cep57, we also performed a live imaging of HeLa cells stably expressing GFP-centrin-1. In control cells, two pairs of engaged centrioles moved towards opposite poles of the cell during prophase–metaphase and lost their connection during late anaphase–cytokinesis. However, Cep57-depleted cells exhibited precocious centriole disengagement already in prophase (66.5 ± 2.1% from three experiments), as observed in fixed cells (Fig. 2d, f and Supplementary Movies 1, 2). Interestingly, precociously disengaged centrioles dynamically moved around in the cell, which sometimes resulted in unequal distribution of centrioles (UDC) in daughter cells (10.0 ± 2.3% from three experiments; Fig. 2e, f, Supplementary Fig. 3c, Supplementary Movie 3). However, most of the Cep57-depleted cells showing the precocious disengagement phenotype still managed to form a bipolar spindle, presumably owing to centriole clustering, and divided into two daughter cells (Supplementary Fig. 3e). Since we noticed that precociously disengaged daughter centrioles aberrantly acquired PCM components such as Cep192, CDK5RAP2, and GTU88 on their surroundings in mitosis, we hypothesized that precociously

disengaged daughter centrioles acquire the ability to nucleate microtubules as MTOCs (Fig. 2a, g, i). In line with this notion, we found that inhibition of centriole clustering by crenolanib[26] in Cep57-depleted cells induced multipolar spindles much more frequently (Supplementary Fig. 3d, e). Crenolanib was reported to induce multipolar cell division in cancer cells with extra centrosomes by cofilin-mediated disruption of the cortical actin cytoskeleton[26]. Moreover, microtubule regrowth assay by cold treatment demonstrated that such precociously matured daughter centrioles obtained ectopic MTOC activity during mitosis (Fig. 2h, i). These data suggest that Cep57 is critical for PCM integrity and centriole engagement, and also that ectopic MTOC activity of disengaged daughter centrioles, provoked by Cep57 depletion, may be deleterious to cells.

**Depletion of Cep57 or *cep57* mutations in MVA disease patients cause chromosome segregation errors and aneuploidy**. Given the above, we next examined the impact of Cep57 depletion on chromosome segregation in HeLa cells. Live-cell imaging showed that Cep57-depleted cells exhibited chromosomal segregation errors such as chromosome misalignment, chromosome lagging, and tripolar spindle formation (abnormal chromosome segregation 12.6%, compared to 2.6% in control cells, from three experiments), and also multi-nuclei formation (Fig. 3a, b, Supplementary Movies 4–6). Similar defects in chromosome segregation were observed in Cep57-depleted fixed cells (Supplementary Fig. 4a, b). We noticed that, in the case of tripolar spindle formation, there were two types of mitotic cells with four or six centrioles (Supplementary Fig. 4a). We assume that a tripolar spindle with six centrioles could arise from the occurrence of UDC and subsequent centriole duplication (Supplementary Fig. 4a). On the other hand, a tripolar spindle with four centrioles was formed because of ectopic MTOC that the precociously disengaged daughter centriole assembled during mitosis (Supplementary Fig. 4a). Moreover, chromosome misalignment, one of the major defects provoked by Cep57 depletion, was efficiently rescued by the expression of Cep57 (Supplementary Fig. 4c). Taken together, Cep57 is crucial for genomic stability by governing PCM integrity and centriole engagement during mitosis.

A previous report showed that kinetochore-localized Cep57 is involved in spindle assembly checkpoint (SAC) and also that depletion of Cep57 causes an impairment of SAC function, and resulting premature onset of anaphase[21]. To judge whether such SAC impairment could cause chromosomal segregation errors, we quantified the time from nuclear envelope breakdown to anaphase onset (AO) upon depletion of Cep57. However, contrary to the previous report[21], we could not observe a

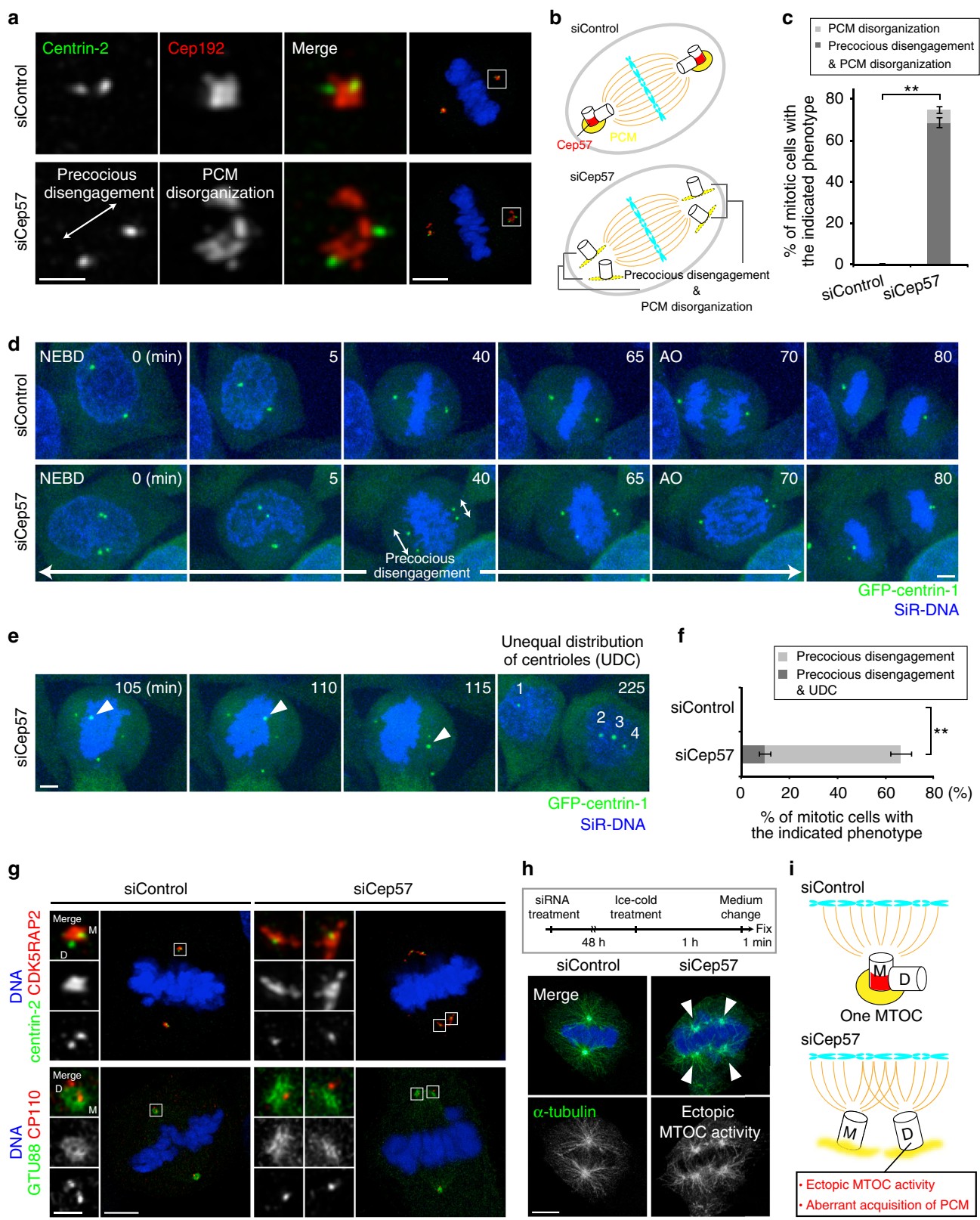

premature metaphase–anaphase transition and SAC deficiency in Cep57-depleted cells, even though we could observe kinetochore signals of Cep57 in control cells in a certain condition (Fig. 3c, Supplementary Fig. 4d). We rather observe a delay in anaphase onset time in Cep57-depleted cells exhibiting chromosome misalignment or multipolar spindles because of ectopic MTOCs (Supplementary Fig. 4e). To further address the effect of Cep57 on the SAC pathway, we measured mitotic duration under the

**Fig. 2** Cep57 is essential for centriole engagement and PCM organization in mitosis. **a** HeLa cells were treated with siControl or siCep57 and immunostained with antibodies against centrin-2 (green) and Cep192 (red). Left–right arrows indicate precociously disengaged centrioles. **b** Schematic model of the phenotype of Cep57 knockdown observed in **a**. Depletion of Cep57 led to PCM disorganization and precocious centriole disengagement. **c** Histograms represent frequency of mitotic cells with the indicated phenotypes observed in **a**. Values are mean percentages ± s.d. from three independent experiments ($n = 50$ for each experiment). **d** Time-lapse observation of cells upon Cep57 depletion. HeLa cells expressing GFP-centrin-1 were treated with siControl or siCep57 and observed in the presence of SiR-DNA (200 nM). Green and blue represent GFP-centrin-1 and DNA, respectively. Left–right arrows indicate precociously disengaged centrioles. Time zero corresponds to the beginning of nuclear envelope breakdown (NEBD). **e** Time-lapse observation of unequal distribution of centrioles (UDC) in Cep57-depleted cells. Precociously disengaged centrioles move around in the cells, which sometimes results in UDC. Arrowheads indicate the precociously disengaged centriole moving toward the opposite spindle pole. **f** Histograms represent frequency of the cells with the indicated phenotypes observed in **e**. Values are mean percentages ± s.d. from three independent experiments (siControl $n = 116$, siCep57 $n = 119$). **g** Aberrant acquisition of PCM components of precociously disengaged daughter centrioles in Cep57-depleted cells. HeLa cells were immunostained with the indicated antibodies **h** Ectopic MTOC activity of precociously disengaged daughter centrioles in Cep57-depleted cells. HeLa cells were immunostained with antibodies against α-Tubulin (green) and CP110 (red). Arrowheads indicate the aberrant MTOC activity in Cep57-depleted mitotic cells. **i** Schematic illustration of the phenotype of Cep57 knockdown observed in **g** and **h**. Precociously disengaged daughter centrioles aberrantly acquire PCM components and MTOC activity. All scale bars, 5 μm in the low-magnified view, 1 μm in the inset. Two-tailed, unpaired Student's *t*-test was used in **c** and **f** to obtain *p* value. **$p < 0.01$

nocodazole treatment. We found that depletion of Cep57 did not decrease mitotic duration, compared to the control cells, whereas depletion of Mad2 bypassed the SAC pathway and significantly shortened mitotic duration (Supplementary Fig. 4f). Moreover, depletion of Cep57 did not reduce the kinetochore signal of Mad1 in HeLa cells treated with nocodazole and MG132 (Supplementary Fig. 4g, h). We therefore reasoned that chromosomal segregation errors caused by Cep57 depletion in this condition are due to precocious centriole disengagement and PCM disorganization, rather than reduced SAC signaling.

We then tested if our finding in human culture cells is also true in MVA patients with biallelic reduction-of-function *cep57* mutations. To this end, we analyzed mitotic features of lymphoblastoid cell lines (LCLs) from MVA patients (Fig. 3d–f, Supplementary Fig. 5a–c). As expected, the MVA patients' LCLs also exhibited precocious centriole disengagement ($40.0 ± 7.2\%$ and $46.7 ± 7.2\%$ in MVA1 and 2 patient cells, respectively) and PCM disorganization ($17.8 ± 1.6\%$ and $20.0 ± 9.8\%$ in MVA1 and 2 patient cells, respectively) in mitosis (from three experiments, Fig. 3e, f, Supplementary Fig. 5d). We also revealed that the number of Cep192 foci was often decreased or increased in MVA patients' interphase cells, compared to control interphase cells that had two Cep192 foci (Fig. 3g, h, Supplementary Fig. 5d). This could be due to UDC as was the case of Cep57-depleted cells. Overall, these findings strongly suggest that both Cep57 depletion and *cep57* mutations in MVA patients cause PCM disorganization and precocious centriole disengagement in mitosis, which results in chromosomal segregation errors and aneuploidy (Fig. 3i).

**Cep57 is an anchor of the PACT domain of PCNT**. To understand how Cep57 contributes to PCM organization and centriole engagement, we then searched for proteins directly binding to Cep57. Given that the cleavage of PCNT, a major PCM component, by separase is required for mitotic centriole disengagement[9,10], and also the fact that, among major PCM components that we tested, depletion of PCNT only caused precocious centriole disengagement in early mitosis (Supplementary Fig. 6a, b), we examined whether Cep57 physically interacts with PCNT. Remarkably, co-immunoprecipitation (co-IP) assay in human HEK293T cells revealed this to be indeed the case (Fig. 4b). Using co-IP assay with deletion constructs of Cep57, we narrowed down the Cep57 region required for PCNT-binding to the short stretch spanning a.a. 68–103 (Fig. 4a, c). Intriguingly, it turns out that this binding region contains the conserved PINC motif of Cep57, suggesting the importance of this interaction in centrosome biogenesis of various species. We

next sought to identify the region of PCNT for binding to Cep57. First, by using co-IP analyses with PCNT truncate mutants, we found that short fragments of PCNT (a.a. 3139–3336 and a.a. 2945–3216) were capable of binding to Cep57 (Fig. 4d, e). These fragments contain the well-conserved PCNT/AKAP9 centrosomal targeting (PACT) domain of PCNT[27]. In contrast, the C-terminal fragment lacking the region of a.a. 3139–3216 (PACT domain) failed to interact with Cep57 (Fig. 5a, b). Since AKAP9 (also known as CG-NAP) also has the PACT domain, we then tested whether Cep57 interacts with the PACT domain of AKAP9. Co-IP analysis with the AKAP9 C-terminus (a.a. 3643–3907) containing the PACT domain showed that Cep57 interacts with the PACT-containing short fragment of AKAP9 (Supplementary Fig. 6c). Second, to confirm the interaction in vitro with the bacterially purified recombinant proteins, we purified Cep57 and PCNT fragments containing the interaction regions that were identified by co-IP experiments. We demonstrated that the two proteins interacted with each other by pull-down assays, suggesting the physical association between Cep57 and PCNT (Fig. 4f, Supplementary Fig. 6d). We noted that the C-terminus PCNT recombinant protein lacking the region of a.a. 3139–3216 (PACT domain) failed to interact with recombinant HA-Cep57, consistent with the binding data in cells (Supplementary Fig. 6d). Pull-down assays also showed the direct interaction between Cep57 and AKAP9 (Supplementary Fig. 6e). We also found that the Cep57–PCNT interaction was reproducibly detected by yeast two-hybrid assay (Supplementary Fig. 6f). Third, to test whether Cep57 is a target of the PACT domain, we co-expressed HA-Cep57 and the GFP-PCNT C-terminus (a.a. 3113–3336) containing the PACT domain in human cells. The GFP-PCNT C-terminus, on its own, localized to centrioles and in the cytoplasm. In contrast, when the GFP-PCNT C-terminus was co-expressed with HA-Cep57, GFP-PCNT C-terminus co-localized with overexpressed Cep57 on microtubules as well as at centrioles (Fig. 4g). These data suggest that the PCNT PACT domain is loaded onto microtubules due to its binding to overexpressed Cep57 that ectopically localizes to the microtubule network. As was the case of the GFP-PCNT C-terminus, the GFP-AKAP9 C-terminus (a.a. 3643–3907) also co-localized with overexpressed Cep57 on microtubules (Fig. 4g). Conversely, we tested whether loss of Cep57 affects centrosomal loading of the PACT. To address this, we overexpressed a GFP-PCNT fragment (a.a. 3132–3226) containing the PACT domain in Cep57-depleted cells, and found that the GFP signal intensity at centrosomes was drastically decreased in response to Cep57 reduction (Fig. 4h, i). Finally, we performed STED analysis to precisely test the co-localization of Cep57 and PCNT around centrioles. STED analysis with N- and C-terminal PCNT antibodies showed that

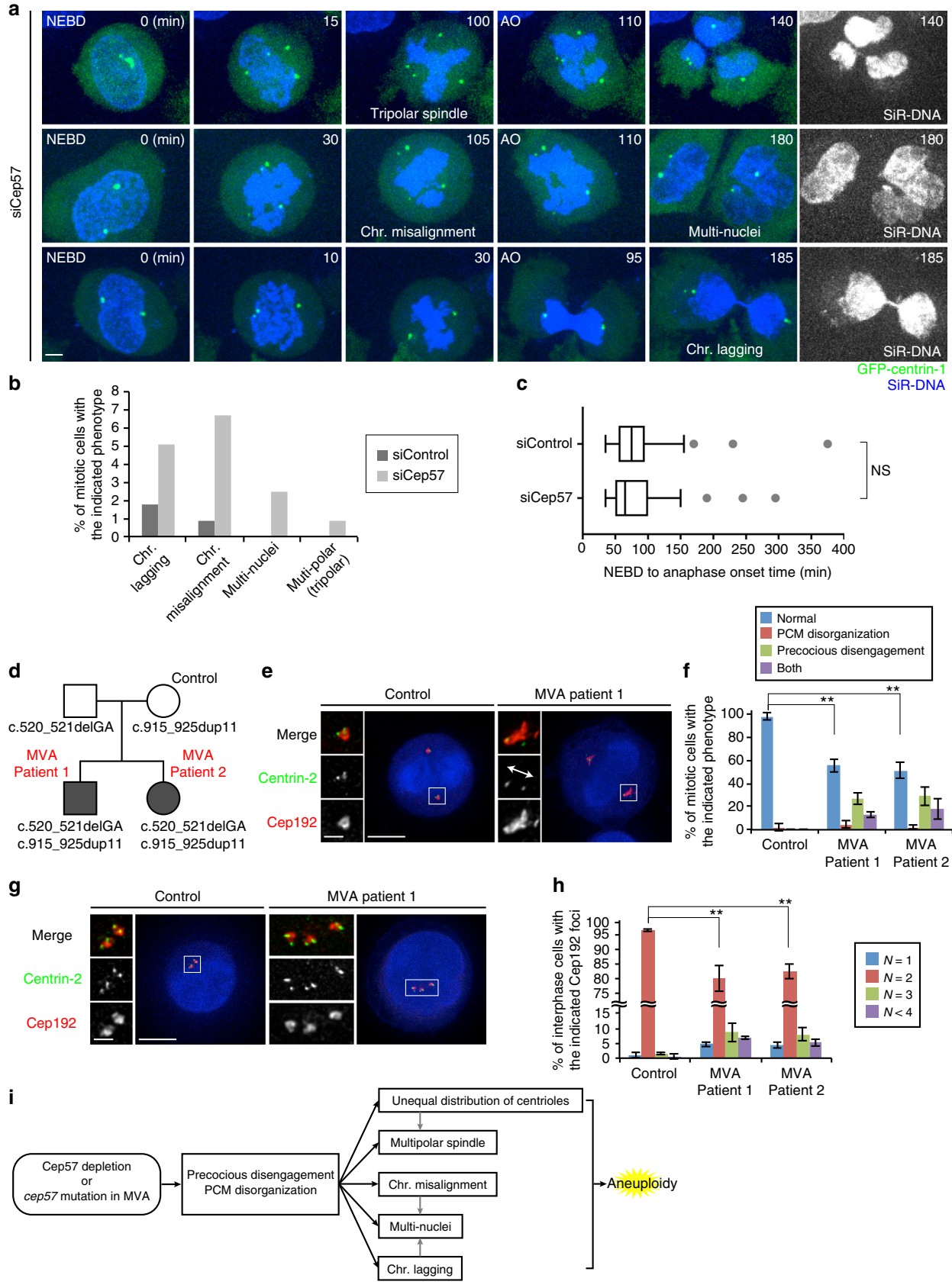

**Fig. 3** Depletion of Cep57 or *cep57* mutations in MVA disease patients cause chromosome segregation errors and aneuploidy. **a** Chromosomal segregation errors observed in Cep57-depleted cells. HeLa GFP-centrin-1 cells were treated with siControl or siCep57 in the presence of SiR-DNA (200 nM). **b** Histograms represent frequency of the mitotic cells with the indicated phenotypes in **a** (siControl *n* = 116, siCep57 *n* = 119 from three independent experiments). **c** Box-and-whiskers plots show the duration from NEBD to AO in **a**. The ends of the box are the upper and lower quartiles. The median is marked by a vertical line inside the box (two independent experiments, siControl *n* = 86, siCep57 *n* = 79). **d** Pedigree of an MVA disease family. MVA patients 1 and 2 were affected with compound heterozygous *cep57* mutations, c.520_521delGA (p.E174fs) and c.915_925dup11 (p.H317fs). Their unaffected father and mother were the carriers of c.520_521delGA and c. 915_925dup11, respectively. **e** MVA patients' cells exhibited precocious centriole disengagement and PCM disorganization. MVA patients' lymphoblastoid cell lines (LCLs) and their mother's LCL (unaffected control) were immunostained with antibodies against centrin-2 (green) and Cep192 (red). Left–right arrows indicate precociously disengaged centrioles. **f** Histograms represent frequency of the control and MVA patients' cells with the indicated phenotypes in **e**. Values are mean percentages ± s.d. from three independent experiments (*n* = 30 for each experiment). **g** Abnormalities in centriole/centrosome number in MVA patients' cells. MVA patients' LCLs and their mother's LCL were immunostained as in **e**. **h** Histograms represent frequency of the control and MVA patients' cells with the indicated number of Cep192 foci in **g**. Values are mean percentages ± s.d. from three independent experiments (*n* = 100 for each experiment). **i** A model for the MVA disease with *cep57* mutations. Depletion of Cep57 or *cep57* mutations in MVA patients cause PCM disorganization and precocious centriole disengagement and thereby result in chromosomal segregation errors and aneuploidy. All scale bars, 5 μm in the low-magnified view, 1 μm in the inset. Mann–Whitney test was used in **c** to obtain *p* value. Dunnett's multiple comparisons test was used in **f** and **h** to obtain *p* value. **\*\****p* < 0.01; NS not significantly different (*p* > 0.05)

Cep57 co-localized with the C-terminus of PCNT, but not with the N-terminus of PCNT in interphase (Supplementary Fig. 6g). Similar observations were true for mitotic cells, and the C-terminus of PCNT co-localized with Cep57 in mitosis although the PCNT signal in mitosis is a bit fuzzy compared to that in interphase, probably due to PCM expansion. Collectively, these data demonstrate that Cep57 acts as an anchor for the PACT domain at centrioles.

Since calmodulin is suggested to be an essential cofactor of the PACT domain in *Drosophila*[28], we tested the effect of calmodulin on the PCM organization in human cells. We did not find any significant defects in loading of GFP-PCNT fragment containing the PACT domain (a.a. 3132–3226) upon calmodulin depletion by two different siRNAs (Supplementary Fig. 7a, b). However, given the observation that addition of calmodulin appeared to modestly promote the Cep57–PCNT interaction at least in vitro (Supplementary Fig. 7c), we cannot officially rule out the possibility that calmodulin somehow facilitates the interaction for proper PCM arrangement.

Although we show that Cep57 is the direct interactor of the PACT domain of PCNT, Cep57 depletion slightly affected the loading of PCNT in interphase (Supplementary Fig. 8a, b). We then assumed that the loading of PCNT is redundantly regulated in interphase. Indeed, we found the PCNT ΔPACT mutant which lacks Cep57-binding region still localized to centrioles through its C-terminus (Supplementary Fig. 8c). We next searched for a new interactor responsible for the loading of PCNT in interphase. Importantly, we found that Cep152 interacted with the C-terminus of PCNT (but not with the PACT domain) and also that CRISPR/Cas9-mediated depletion of Cep152 significantly decreased the centriolar signal intensity of PCNT in interphase (Supplementary Fig. 8d–f). Overexpression of CRISPR/Cas9-resistant GFP-Cep152 restored the PCNT signal at centrioles in the same condition, confirming the specificity of Cep152 guideRNA (Supplementary Fig. 8e, f). These data indicate that Cep57 and Cep152, and perhaps other factors, redundantly regulate the loading of PCNT in interphase.

**The binding of PCNT to Cep57 at centrioles is required for the proper localization pattern of PCNT to organize mitotic PCM.** Importantly, we found that two mutations in the *PCNT* gene, K3154del mutation within the PACT domain and R2918X truncate mutation, which are responsible for microcephalic osteodysplastic primordial dwarfism type 2 (MOPD2) disease[29,30], drastically reduced the binding to Cep57, compared to wild-type PCNT (Fig. 5a–c). This observation suggests that the disruption of the Cep57–PCNT interaction might be a cause of

the MOPD2 disease, though we cannot officially rule out the possibility that the mutations also disrupt another protein–protein interaction. We next set out to examine the biological significance of the Cep57–PCNT interaction for PCM integrity and centriole engagement. To this end, we depleted endogenous PCNT using siRNA and expressed RNAi-resistant (RNAi-R) full-length or Cep57-binding-deficient mutants (PCNT ΔPACT or K3154del). All forms of PCNT were normally recruited to the centrosome in interphase (Supplementary Fig. 9a). Whereas depletion of endogenous PCNT caused precocious centriole disengagement in mitosis, expression of the full-length PCNT functionally rescued this phenotype (Fig. 5d, e). In contrast, the two binding-deficient mutants failed to rescue the phenotype provoked by depletion of endogenous PCNT (Fig. 5d, e). Notably, these two mutants could not form PCM normally and caused PCM disorganization as observed in Cep57-depleted culture cells. These data suggest that the binding of PCNT to Cep57 at centrioles is required for the proper localization pattern of PCNT to organize the highly-ordered PCM structure and to maintain the centriole engagement during mitosis. In line with this notion, we established that depletion of Cep57 caused PCNT dispersion (Supplementary Fig. 9b) and significantly shortened the height of PCNT that binds along the side of centriolar wall (Supplementary Fig. 9c, d). Similarly, we sought to perform Cep57 knockdown-rescue experiments with Cep57 deletion mutant lacking the PCNT-binding domain; however, the PCNT-binding domain of Cep57 overlaps with its centriolar localization domain and the Cep57 mutant failed to localize to centrioles (Fig. 4c, Supplementary Fig. 1i). As expected, we demonstrated that expression of the Cep57 mutant did not rescue the phenotype provoked by Cep57 depletion (Supplementary Fig. 9e).

To further confirm the significance of the interaction, we next designed a PCNT chimera where the PACT domain was replaced with Cep57 (PCNT-Cep57 chimera) and performed PCNT knockdown-rescue experiments with the PCNT-Cep57 chimera construct (Fig. 5f). Interestingly, this chimera mutant efficiently rescued the disengagement phenotype of PCNT depletion (Fig. 5g, h). These data indicate that the Cep57–PCNT interaction is crucial for the function of PCNT at centrosomes and mitotic PCM organization (Fig. 6).

## Discussion
In this study, we addressed the role of Cep57 in centriole engagement and PCM organization. We revealed that Cep57 localizes at the vicinity of centrioles and that Cep57 depletion resulted in precocious centriole disengagement and PCM disorganization in mitosis (Fig. 6). Interestingly, such precociously

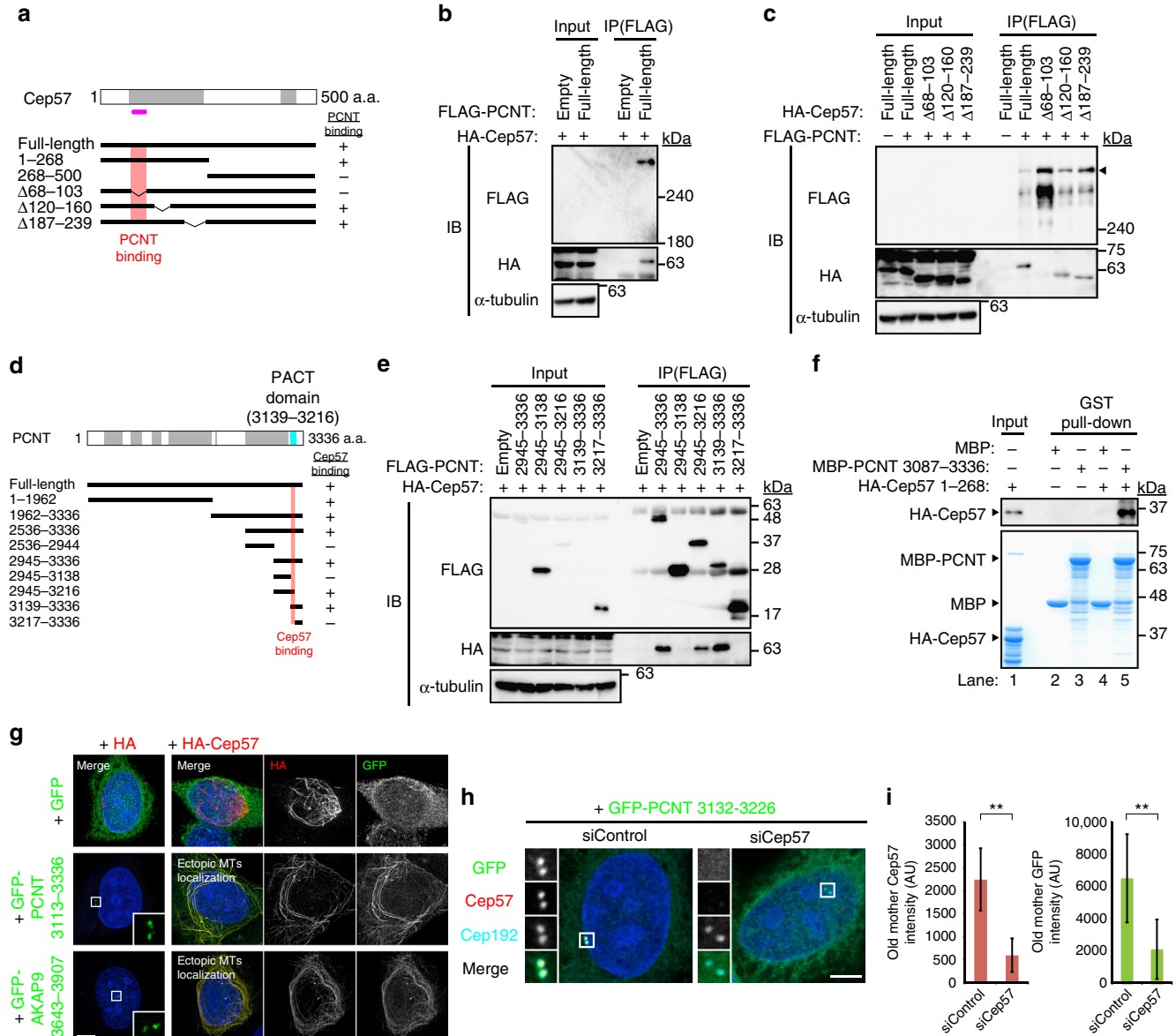

**Fig. 4** Cep57 is an anchor of the PACT domain of PCNT. **a** Schematic of Cep57 and the deletion mutants used for co-IP assays. Coiled-coil domains are shown in gray box; the PINC motif in pink line. The region of Cep57 for PCNT-binding is represented in red. The right column shows a summary of the co-IP results. **b** HEK293T cells co-expressing FLAG empty (control) or FLAG-PCNT and HA-Cep57 were immunoprecipitated (IPed) with FLAG antibodies. **c** HEK293T cells co-expressing FLAG-PCNT and HA-Cep57 or the indicated deletion mutants were IPed with FLAG antibodies. **d** Schematic of PCNT and the deletion mutants used for co-IP assays. Coiled-coil domains are shown in gray box, and the PACT domain in blue box. The region of PCNT for Cep57-binding is represented in red. The right column shows a summary of the co-IP results. We noticed that the N-terminal half of PCNT (a.a. 1–1962), which does not contain the PACT domain, also interacted with Cep57. **e** HEK293T cells co-expressing HA-Cep57 and FLAG-PCNT or the indicated deletion mutants were IPed with FLAG antibodies. **f** MBP pull-down assay showing the interaction between Cep57 and PCNT fragments in vitro. These bacterially purified recombinant proteins contain the interaction regions that were identified by the co-IP experiments in **c** and **e**. Inputs (lane 1 for Coomassie blue staining, 13.3%; lane 1 for western blotting, 1/30,000 volume of lane 1 in the Coomassie blue staining) and affinity-purified protein complexes (lanes 2–5) were subjected to SDS-PAGE, stained (Coomassie blue staining), and analyzed by western blotting. **g** HeLa cells co-expressing HA empty (control) or HA-Cep57 and GFP or GFP-PCNT 3113–3336 were immunostained with antibodies against HA (red), GFP (green). **h** HeLa cells were treated with control siRNA or siCep57, followed by transfection with the GFP-PCNT 3132–3226. The cells were immunostained with antibodies against GFP (green), Cep57 (red), and Cep192 (cyan). **i** Histograms represent quantification of the signal intensity of GFP and Cep57 at old mother centrioles in **h** (siControl n = 41, siCep57 n = 39). All scale bars, 5 µm. **p < 0.01

disengaged daughter centrioles aberrantly recruit PCM components and acquire ectopic MTOC activity, which thereby appears to cause chromosome segregation errors and aneuploidy in human cells. Moreover, our study provides evidence that precociously disengaged centrioles can be unequally distributed into two daughter cells, which also leads to chromosome segregation errors and aneuploidy. Importantly, MVA patients' cells also exhibit similar defects, such as precocious centriole disengagement and UDC, suggesting a potential cause of the MVA disease.

The PACT domain has been utilized for targeting a protein of interest to centrioles in the fields since it was discovered in 2000 (ref. [27]). However, the exact binding partner of the PACT domain has not yet been determined. Our study showed that Cep57 directly binds to the PACT domains of PCNT and AKAP9, and

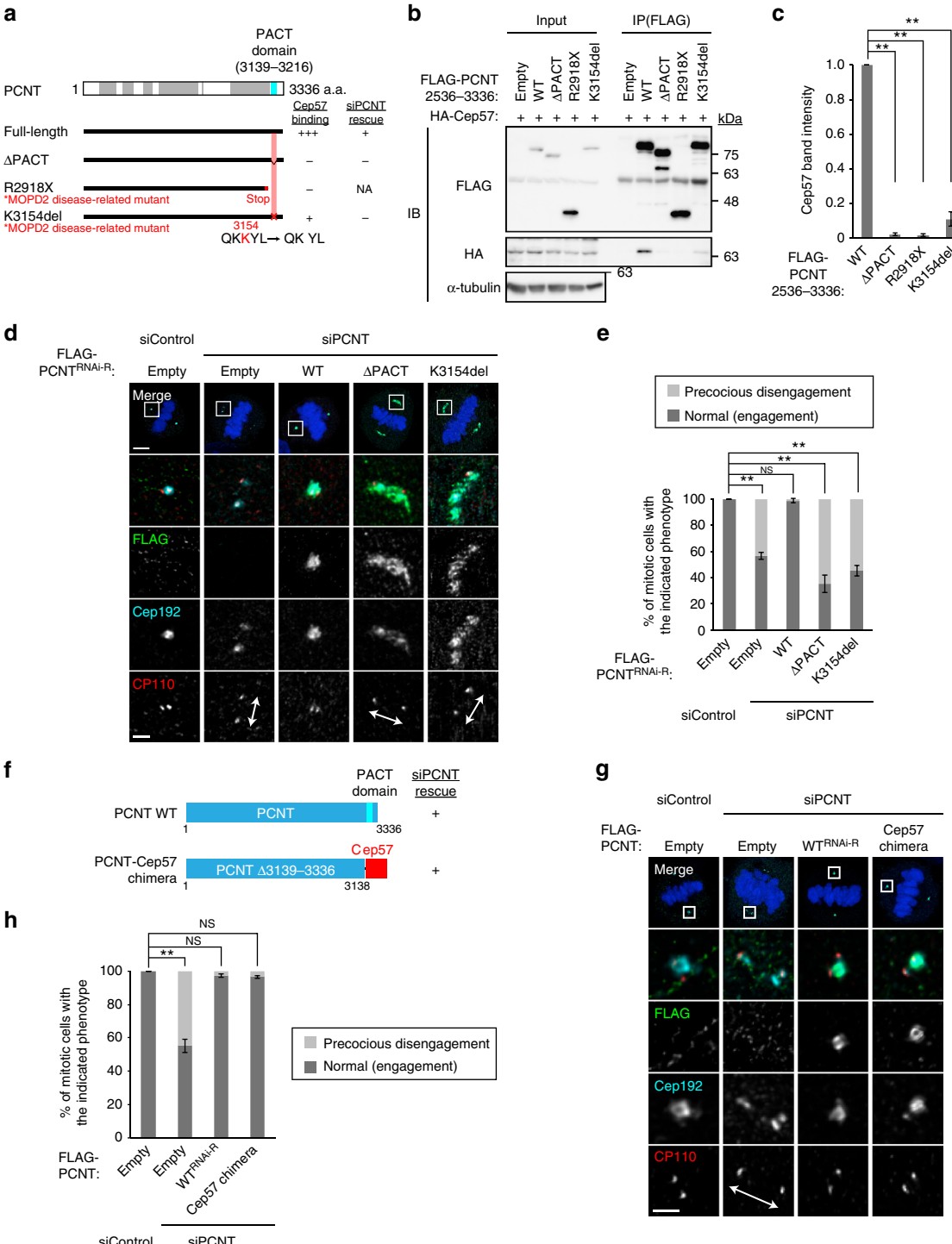

that the binding of the PCNT PACT domain to Cep57 is critical for proper arrangement of PCNT in centrosomes to organize the functional PCM structure. Importantly, MOPD2 disease-related *PCNT* mutations impair the Cep57–PCNT interaction and phenocopy the mitotic phenotypes provoked by depletion of endogenous Cep57, suggesting a potential cause of the MOPD2 disease. Overall, these findings led us to propose that the Cep57–PCNT module provides a critical interphase between the centriole core and PCM.

Unlike mitosis, however, depletion of Cep57 or PCNT did not affect PCM organization and centriole engagement in interphase, although both Cep57 and PCNT co-localized at the vicinity of mother centrioles from the G1 phase. We therefore assume that a redundant mechanism somehow maintains centriole engagement in interphase, independently of the Cep57–Pericentrin interaction. Based on this, it would be possible that the PCM expansion involves overall rearrangement of PCM components so as to couple the expansion process with maintenance of centriole engagement during mitosis, rather than simple accumulation of additional PCM components on top of the pre-existing PCM layer. In line with this, we found that inhibition of Plk1, a major kinase for PCM expansion[18], blocked precocious centriole disengagement during mitosis in Cep57-depleted cells (Supplementary Fig. 9f, g). Therefore, unraveling the potential role of Plk1 in regulating the Cep57–PCNT module will be a fascinating topic for future study.

**Fig. 5** The binding of PCNT to Cep57 at centrioles is required for the proper localization pattern of PCNT to organize mitotic PCM. **a** Schematic of PCNT and the ΔPACT mutant or MOPD2 disease-related mutants used for co-IP assays. Coiled-coil domains are shown in gray box and the PACT domain in blue box. The region of PCNT for Cep57-binding is represented in red. The right columns show a summary of the co-IP results and rescue experiments, respectively. **b** HEK293T cells co-expressing HA-Cep57 and FLAG-PCNT full-length or the indicated mutants were IPed with FLAG antibodies. **c** The signal intensity of HA-Cep57 bands in **b** was quantified. Values are mean ± s.d. from three independent experiments. Tukey's multiple comparisons test was used to obtain $p$ value. $*p < 0.05$; $**p < 0.01$; NS not significantly different ($p > 0.05$). **d** HeLa cells were treated with siControl or siPCNT, followed by transfection with FLAG empty (control), RNAi-resistant (RNAi-R) PCNT or the indicated mutants. The cells were immunostained with antibodies against FLAG (green), Cep192 (cyan) and CP110 (red). **e** Histograms represent frequency of mitotic cells with the indicated phenotypes observed in **d** in each condition. Values are mean percentages ± s.d. from three independent experiments ($n = 50$ for each experiment). **f** Schematic of PCNT and the PCNT-Cep57 chimera mutant used for PCNT knockdown and rescue experiments. **g** HeLa cells were treated with siControl or siPCNT, followed by transfection with FLAG empty (control), RNAi-resistant PCNT, or the PCNT-Cep57 chimera mutant. The cells were immunostained with antibodies against FLAG (green), Cep192 (cyan), and CP110 (red). **h** Histograms represent frequency of mitotic cells with the indicated phenotypes observed in **g** in each condition. Values are mean percentages ± s.d. from three independent experiments ($n = 50$ for each experiment). All scale bars, 5 μm in the low-magnified view, 1 μm in the inset. Left–right arrows indicate precociously disengaged centrioles. Tukey's multiple comparisons test was used in **e** and **h** to obtain $p$ value. $*p < 0.05$; $**p < 0.01$; NS not significantly different ($p > 0.05$)

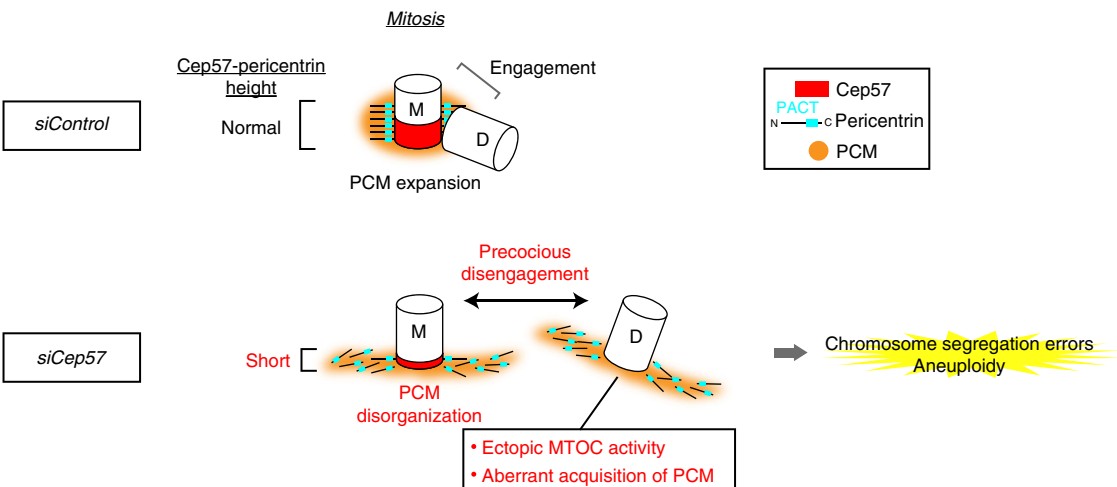

**Fig. 6** A speculative model for the role of Cep57 in centriole engagement and PCM integrity during mitosis. Cep57 localizes at the proximal ends of mother centrioles and directly binds to the PACT domain of PCNT. The binding of PCNT to Cep57 at centrioles is required for the proper localization pattern of PCNT to organize the highly ordered PCM structure and to maintain the centriole engagement during mitosis. In the absence of Cep57, the PCNT height that binds along the side of centriole wall is affected, which causes PCNT dispersion and PCM disorganization. PCM disorganization further leads to precocious centriole disengagement during mitosis. Precociously disengaged daughter centrioles aberrantly recruit PCM components and acquire ectopic MTOC activity, which thereby causes chromosome segregation errors and aneuploidy

Cep57 was previously suggested to be a PCM component and involved in spindle pole focusing[24]. We now reevaluated the precise localization pattern of Cep57 and found that Cep57 localization is limited at the vicinity of centrioles. Although the previous report showed that the PCM fragmentation is a cause of multipolar spindle formation in Cep57-depleted cells[24], we conclude that the precocious centriole disengagement and resulting ectopic MTOC formation are the direct cause of the multipolar spindle formation. In fact, apparent multipolar phenotype is relatively rare, but a pseudo-bipolar spindle with multiple MTOCs is a major phenotype in Cep57-depleted cells, which can induce chromosome segregation errors.

A previous study showed that Ppc89, an *S. pombe* functional homolog of Cep57, is also critical for spindle pole integrity, whereas fission yeast has spindle pole bodies (SPB) instead of centrosomes[31]. In addition, it has recently been suggested that the N-terminus of Ppc89 and the C-terminus of Pcp1 (*S. pombe* homolog of PCNT) are in close proximity as is the case of the N-terminus of Cep57 and the C-terminus of PCNT in human[32]. It is therefore tempting to speculate that Cep57 and Ppc89 provide a similar function serving as a critical interface within the centrosome and SPB, respectively.

## Methods

**Cell culture and transfection.** HeLa, U2OS, RPE-1, and HEK293T cells were obtained from the ECACC (European collection of cell cultures). 293FT were obtained from ThermoFisher. HeLa, U2OS, HEK293T, and 293FT cells were cultured in Dulbecco's modified Eagle's medium (DMEM) supplemented 10% fetal bovine serum (FBS) and 100 μg/ml penicillin–streptomycin at 37 °C in a 5% $CO_2$ atmosphere. RPE-1 cells were cultured in DMEM/F12 medium supplemented 10% FBS and 100 μg/ml penicillin–streptomycin at 37 °C in a 5% $CO_2$ atmosphere. For the cytological analysis, the following MVA patients LCLs were used: Coriell ID_GM21654 (patient 1 with biallelic CEP57 mutations) and Coriell ID_GM21655 (patient 2 with biallelic CEP57 mutations). Control LCLs were also used: Coriell ID_GM21656 (mother of two affected children). The LCLs were cultured in RPMI supplemented 15% FBS, 100 μg/ml penicillin–streptomycin, 2 mM alanyl-glutamine at 37 °C in a 5% $CO_2$ atmosphere.

Transfection of siRNA or DNA constructs into HeLa, U2OS, and HEK293T cells was conducted using Lipofectamine RNAiMAX (Life Technologies) or Lipofectamine 2000 (Life Technologies), respectively. Unless otherwise noted, the transfected cells were analyzed 48–72 h after transfection with siRNA and 24–36 h after transfection with DNA constructs.

**RNA interference.** The following siRNAs were used: Silencer Select siRNA (Life Technologies) against Cep57 #1 (s18692), Cep57 #2 (s18694), Cep57 #3 (s18693), PCNT #1 (s10136), PCNT #2 (s10137), PCNT #3 (s10138), CDK5RAP2 #1 (s31428), CDK5RAP2 #2 (s31429), CDK5RAP2 #3 (s31430), AKAP9 #1 (s19745), AKAP9 #2 (s19746), AKAP9 #3 (s19746), Mad2 (s8391), calmodulin1 #1 (s2340), calmodulin1 #2 (s2341), and negative control #1 (4390843). Unless noted, Cep57 #1 and PCNT #3 were used in this study.

**SgRNA.** HeLa cells stably expressing Cas9 (HeLa-Cas9) were generated by lentivirus infection. For lentivirus infection, pKLV2-EF1a-BsdCas9-W (Addgene #67978) was introduced into 293FT cells together with psPAX2 (Addgene #12260) and pCMV-VSV-G (Addgene #8454). The recombinant retroviruses thereby generated were used to infect HeLa cells, which were then subjected to limited dilution to isolate single colonies. pX330-U6-Chimeric_BB-CBh-hSpCas9 (pX330-hSpCas9) was a gift from Dr. Feng Zhang (Addgene #42230). The hSpCas9 sequences in the original vector were substituted by PuroR coding sequences (pX330-PuroR). Single guideRNA (sgRNA) oligo targeting Cep152 (forward: CACCGAGGACCATGTCATTAGACTT, reverse: AAACAAGTC TAATGACATGGTCCTC) was cloned into the *Bbs*I site of pX330-PuroR. The resulting vector was introduced into HeLa-Cas9 cells using TransIT-2020 (Mirus Bio) and subjected to selection in medium containing puromycin (2 μg/ml).

**Plasmids.** Complementary DNA (cDNA) encoding full-length Cep57 isoform a (NCBI NP_055494.2) was amplified from cDNA library of HeLa cells. The Cep57 cDNA was subcloned into pCMV5-HA (Addgene). pTB701 constructs encoding full-length PCNT were described previously[33]. The Cep57 and PCNT deletion mutant constructs were created using PrimeSTAR mutagenesis basal kit (TaKaRa) and In-Fusion HD cloning kit (Clontech) according to the manufacturer's protocol. For bacterial expression, cDNAs for HA-tagged Cep57, PCNT, or AKAP9 fragments were subcloned into pGEX-6p-1 (GE Healthcare).

**Antibodies.** The following primary antibodies were used in this study: rabbit antibodies against Cep57 (GeneTex, GTX115931, IF 1:1000), PCNT (Abcam, ab4448, IF 1:2000), Cep192 (Bethyl Laboratories, A302–324 A, IF 1:1000), Cep152 (Bethyl Laboratories, A302–480A, IF 1:1000), CP110 (Proteintech, 12780-1-AP, IF 1:500), CDK5RAP2 (Bethyl Laboratories, IHC-0063, IF 1:1000), Mad1 (Bethyl Laboratories, A300-339A, IF 1:500), ODF-2 (Abcam, ab43840, IF 1:1000), and HA-tag (Abcam, ab9110, IF 1:1000, WB 1:1000); mouse antibodies against Cep57 (Abcam, ab169301, IF1:1000), PCNT (Abcam, ab28144, IF 1:2000), centrin-2 (Merck, 20H5, IF 1:1000), γ-tubulin (GTU88) (Merck, T5192, IF 1:1000), poly-glutamylation modification (GT335) (AdipoGen, AG-20B-0020-C100, IF 1:1000), GFP (MBL, 598, IF 1:1000, WB 1:1000), FLAG-tag (Merck, F1804, IF 1:1000, WB 1:1000), and α-tubulin (Merck, DM1A, IF1:1000, WB 1:1000); goat antibodies against PCNT (Santa Cruz Biotechnology, sc-218145, 1:300); guinea pig antibodies against CENP-C (MBL, PD030, IF 1:1000); Alexa 488-labeled Cep152 (Bethyl Laboratories, A302-480A, IF 1:500) and ODF-2 (Abcam, ab43840, IF 1:100) and Alexa 647-labeled Cep192 (Bethyl Laboratories, A302-324A, IF 1:500) were generated with Alexa Fluor labeling kits (Life Technologies). The following secondary antibodies were used: Alexa Fluor 488 goat anti-mouse IgG (H + L) (Molecular Probes, A-11001, 1:1000), Alexa Fluor 568 goat anti-rabbit IgG (H + L) (Molecular Probes, A-11011, 1:1000), Alexa Fluor 568 goat anti-guinea pig IgG (H + L) (Abcam, ab175714, 1:1000), Alexa Fluor 555 donkey anti-rabbit IgG (H + L) (Abcam, ab150074, 1:1000), and Alexa Fluor 488 donkey anti-goat IgG (H + L) (Abcam, ab150129, 1:1000) for IF; Alexa Fluor 555 goat anti-rabbit IgG (H + L) (Molecular Probes, A-21428, 1:500) for STED; Goat polyclonal antibodies-horseradish peroxidase against mouse IgG (Promega, W402B, 1:5000) and rabbit IgG (Promega, W401B, 1:5000) for WB.

**Drug treatment.** The following chemicals were used in this study: crenolanib (Selleck, CP-868596), nocodazole (Wako, 31430-18-9), MG132 (Wako, 133407-82-6), BI2536 (AdooQ, A10134), and SiR-DNA (Spirochrome, CY-SC007).

**Microscopy.** For immunofluorescence analysis, the cells cultured on coverslips (Matsunami, 18 mm, thickness No. 1_0.12-0.17 mm for a confocal microscope; Matsunami, 22 × 22 mm, thickness No. 1s_0.16-0.19 mm for an STED microscope) were fixed using −20 °C methanol for 7 min and washed with PBS. The cells were permeabilized after fixation with PBS/0.05% Triton X-100 (PBSX) for 5 min three times, and incubated for blocking in 1% BSA in PBSX for 30 min at room temperature (RT). The cells were incubated with primary antibodies for 24 h at 4 °C, washed with PBSX three times, and incubated with secondary antibodies for 1 h at RT. The cells were thereafter washed with PBSX twice, stained with 0.2 μg/ml Hoechst 33258 (DOJINDO) in PBS for 5 min at RT, washed again with PBSX, and mounted onto glass slides.

Counting the number of immunofluorescence signals was preformed using an Axioplan2 fluorescence microscope (Carl Zeiss) with a ×60 or ×100/1.4 NA plan-APOCHROMAT objective and DeltaVision Personal DV-SoftWoRx system (Applied Precision) equipped with a CoolSNAP CH350 CCD camera.

Confocal microscopy images were taken by the Leica TCS SP8 HSR system equipped with a Leica HCX PL APO ×63/1.4 oil CS2 objectives and excitation wavelength 405, 488 and 561 nm. To obtain high-resolution images, the pinhole was adjusted at 0.5 airy units. Scan speed was set to 200 Hz in combination with five-fold line average in 856 × 100 format (pixel size 43 nm) or four-fold line average in 1080 × 500 format (pixel size 43 nm). The images were collected at 130 nm z steps. For deconvolution, Huygens essential software (SVI; Scientific Volume Imaging) was used.

STED images were taken by a Leica TCS SP8 STED 3X system with a Leica HC PL APO ×100/1.40 oil STED WHITE, and 660 nm gated STED. Scan speed was set to 100 Hz in combination with five-fold line average in a 512 × 80 format (pixel size

15–20 nm). The images were collected at 180 nm z steps. The STED images were processed by deconvolution with Huygens professional software (SVI). The resolution of green signal (~80 nm; Alexa488) is generally lower than that of red signal (~50 nm; Alexa555) in this system. In this study, we mainly looked at the proteins of our interest at centrioles in red whereas the other protein was visualized in green.

**Diameter determination of the Cep57 ring.** The ring diameters of Cep57 and Cep192 were measured using the ImageJ plugin "Radial Profile Extended". The plugin was run with a circular ROI slightly bigger than the Cep57/Cep192 ring so that a radial profile from the center of the ring was obtained as in Fig. 1e. The obtained profile was then fitted with a Gaussian curve using Mathematica and the distance between the center of the ring and the peak of the Gaussian curve was defined as the radius.

**Live imaging.** A Confocal Scanner Box, Cell Voyager CV1000 (Yokogawa Electric Corp.) equipped with a ×63 oil immersion objective lens and the stage incubator for a 35-mm dish was used for live cell imaging. HeLa cells stably expressing GFP-centrin-1 were cultured on 35 mm glass-bottom dishes (Greiner-bio-one, #627870) at 37 °C in a 5% CO₂ atmosphere. Before imaging, cells were treated with siRNAs for 24 h and with 200 nM of SiR-DNA for 6 h. Images were taken by a Back-illuminated EMCCD camera. After addition of drugs or after 24-48 h from transfection, the cells were visualized every 5 min over 24 h. The images were collected at 1.2 μm z steps. Maximum projections were generated using ImageJ (National Institutes of Health).

**Immunoprecipitation and western blotting.** For preparation of human cell lysates for western blotting, HEK293T cells were collected 24–36 h after transfection, lysed on ice in lysis buffer (20 mM Tris/HCl pH 7.5, 50 mM NaCl, 1% TritonX-100, 5 mM EGTA, 1 mM DTT, 2 mM MgCl₂ and 1/1000 protease inhibitor cocktail (Nakarai tesque)). Insoluble material was removed after centrifugation for 10 min. For IP of FLAG-tagged proteins, whole-cell lysates were incubated with FLAG antibody-conjugated M2 agarose gel (Merck) for 2 h at 4 °C. The beads were washed at least three times with lysis buffer and resuspended in sodium dodecyl sulfate (SDS) sample buffer before loading onto 6–15% polyacrylamide gels, followed by transfer on Immobilon-P membrane (Merk). The membrane was probed with the primary antibodies, followed by incubation with their respective horseradish peroxidase-conjugated secondary antibodies (Promega). Washes were performed in PBS containing 0.02% Tween (PBST). The signal was detected with Chemi Doc XRS + (BIO RAD). Signal intensity of immuno-reactive bands was measured using Adobe Photoshop. Unless otherwise specified, the experiments of western blotting were repeated at least three times. The antibody against α-tubulin was used as a loading control.

**In vitro binding assay.** Cep57 or PCNT fragments were cloned in pGEX system vectors (GE Healthcare) encoding for GST- or MBP-tags. These analyses were performed as described previously[34] with slight modifications. The recombinant protein expression of the fragments was performed in *E. coli* strain BL21 gold (DE3) in LB medium. Protein expression was induced at 18 °C by addition of 0.3 mM IPTG and allowed to proceed for 18 h. Cell pellets expressing GST-PCNT, MBP-PCNT, or HA-Cep57 fragments were lysed by lysozyme treatment and sonication, resuspended in lysis buffer containing 50 mM Tris-HCl (pH 7.5), 500 mM NaCl, 5 mM EDTA, 1 mM DTT, 1:1000 protease inhibitor cocktail (Merck), and 0.5% TritonX-100. The lysates were incubated with glutathione sepharose beads (GE Healthcare) or amylose resin (New England BioLabs). The beads were then washed ten times with lysis buffer supplemented with additional 500 mM NaCl at high salt concentrations. Cep57 fragment proteins were then eluted from the beads by removal of the GST-tags by prescission protease (GE Healthcare) in a buffer containing 20 mM Tris-HCl (pH 7.5), 150 mM NaCl, 0.5 mM EDTA, 1 mM DTT. Recombinant calmodulin 1 protein (Merck) was commercially available.

For GST pull-down assay, the glutathione sepharose beads which retain bacterially purified GST-PCNT fragment were resuspended in ice-cold lysis buffer and incubated with ~5 μg bacterially purified HA-Cep57 for 2–4 h at 4 °C. The beads were washed three times with ice-cold lysis buffer and resuspended in SDS sample buffer. For MBP pull-down assay, the amylose resin which retains bacterially purified MBP-PCNT fragment was resuspended in ice-cold lysis buffer and incubated with ~5 μg bacterially purified HA-Cep57 (and/or calmodulin 1) for 2–4 h at 4 °C. The resin was washed three times with ice-cold lysis buffer and resuspended in SDS sample buffer. Proteins were separated by SDS polyacrylamide gel electrophoresis and stained with SimplyBlue Safe (Invitrogen).

**Yeast two-hybrid analysis.** Fragments of Cep57 or PCNT were cloned into the vectors pSM671 (bait) and pSM378 (prey) (gifts from Satoru Mimura). Yeast strain L40 was grown in complete medium (yeast extract peptone dextrose; (YPD)) and transformed with the indicated vectors. Positive colonies were grown on plates lacking leucine and tryptophan in the presence of histidine at 30 °C. After a few days, cells were streaked on plates containing 20 mM 3-amino-1,2,4-triazole (Merck) without leucine, tryptophan, and histidine. The plates were incubated at 30 °C for a few days. The same results were obtained from two independent clones for each combination.

**Reverse transcription and real-time PCR**. Total RNA (0.5 μg) isolated from cells with the use of TRIZOL reagent (Thermo Fischer Scientific) was subject to reverse transcription (RT) with ReverTra Ace (Toyobo), and the resulting cDNA was subjected to real-time PCR analysis with SYBR Green PCR Master Mix (TaKaRa) and specific primers in a StepOnePlus Real-Time PCR system (Applied Biosystems). The sequence of the primers (sense and antisense, respectively) were 5′-aagcctacacttgcctatccag-3′ and 5′-ttccaagcgtcgaatcttatc-3′ for human Cep57, and 5′-tccactggcgtccttcacc-3′ and 5′-ggcagagatgatgaccctttt-3′ for human GAPDH. The amount of Cep57 mRNA was normalized by that of GAPDH mRNA.

**Reporting Summary**. Further information on experimental design is available in the Nature Research Reporting Summary linked to this article.

## Data availability

The data that support the findings of this study are available from the corresponding author upon request.

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

## Acknowledgements

We thank Y. Nozaki and T. Ashikawa for supporting the experiments, as well as the Kitagawa lab members for fruitful discussion. This work was supported by SOKENDAI (The Graduate University for Advanced Studies), by Grant-in-Aid for Young Scientists (A) and for JSPS Fellows and for Scientific Research on Innovative Areas from the Ministry of Education, Science, Sports and Culture of Japan, by Takeda Science Foundation, by Daiichi Sankyo foundation of Life Science, by Mochida Memorial Foundation for Medical and Pharmaceutical Research, and by the Uehara Memorial Foundation.

## Author contributions

K.W. and D.K. designed the study; K.W. and K.I. performed experiments; K.W., M.T. and D.K. designed experiments; K.W. and D.T. analyzed data; K.W. and D.K. wrote the manuscript, which was commented on by all authors.

## Additional information

**Competing interests:** The authors declare no competing interests.

