## [Peer Review File · Nature Communications]

Reviewers' comments:

Reviewer #1 (Remarks to the Author):

In this manuscript, Kitagawa and his colleagues reported roles of Cep57 in the centrosome. They revealed that Cep57 is located at the proximal end of mother centriole. Depletion of Cep57 causes PCM disorganization and precocious centriole disengagement during mitosis. As a result, the disengaged daughter centrioles function as spindle poles, leading to chromosome mis-segregation and aneuploidy. Similar defects were observed in mosaic variegated aneuploidy syndrome patient cells with cep57 mutations. They also observed a specific interaction of Cep57 with the PACT domain of pericentrin. Based on the results, the authors proposed that the Cep57-pericentrin complex provides a critical interphase between centriole core and PCM, which is crucial for the higher-order structure of the PCM.

This work revealed how Cep57 works at the centrosome. These results are novel and potentially important for elucidating structural and functional linkage between centrioles and PCM. Overall, their experiments seemed to be carefully carried out and the results looked trustful. This work is worth to be published in Nature Communication. The follows are major points to be resolved prior to publication.

1. The authors may have to explain why their results are different from the previous papers in which Cep57 functions in the kinetochore. One of critical differences between this work and Zhou et al. (2016) may be the immunostaining of Cep57 in mitotic cells. The Cep57 signals were coimmunostained with CREST, CENP-A and Mis12 in Zhou et al. (2016). On the other hand, this manuscript did not show an immunostaining image of the whole mitotic cells with the Cep57 antibody.

2. To enforce their proposal, knockdown-rescue experiments of Fig. 2f,g may be performed with HA-Cep57 deletion ($\Delta 68-103$) and truncated mutants (1-530; Zhou et al., 2016).

3. The authors reported that depletion of Cep57 did not lead to any defects in PCM formation and centriole duplication in interphase cells. It is worth to determine that depletion of Cep57 hardly affect daughter centriole recruitment of Cep152 in interphase cells.

4. Precocious centriole disengagement was observed in Cep57-depleted cells (Figs. 3a). The separated centrioles with ectopic MTOC activities eventually became clustered to form pseudo-bipoles (Fig. 2h, Supplementary Fig. 2i,j). Nonetheless, the authors did not observe a delay in NEBD to anaphase onset time in Cep57-depleted cells (Fig. 3c). In fact, Zhou et al. (2016) reported a shortening of the NEBD to anaphase onset time in Cep57-depleted cells. The authors need to explain this observation, based on your proposal.

5. The model proposes that pericentrin interacts with Cep57 at the mother centriole. This configuration may be acceptable only in interphase cells. If the model is correct, then precocious centriole disengagement should be observed in interphase cells. According to Lawo et al. (2012), the PCM structure of the mitotic cells are quite different with massive accumulation of pericentrin and other components. Therefore, the model should be corrected for mitotic cells.

6. The authors need to define how to discriminate precocious centriole disengagement.

Reviewer #2 (Remarks to the Author):

The manuscript by Watanabe and colleagues explores the role of Cep57 in centriole engagement. They show that upon Cep57 depletion the mother-daughter centriole prematurely disengage as

early as prophase, when centrioles normally disengage in late mitosis. The authors show that early disengagement leads to disorganized PCM, splayed spindle poles, and ultimately defects in chromosome segregation. Given that Pericentrin also shows a similar centriole disengagement and PCM phenotypes upon depletion, the authors explore the possibility that Cep57 and Pericentrin function together. They indeed show direct binding of these two proteins that depends on the newly identified and conserved PINC domain in Cep57 and the PACT domain in Pericentrin. The authors also suggest that the disease linked mutation K3154del in Pericentrin might cause the disease as a result of loss of interaction with Cep57.

This study is potentially interesting as it more clearly explains the previously shown multipolar spindle phenotype by identifying a centriole disengagement phenotype. As you will see below, the authors must do additional experiments to ensure that the two distant centrioles at the pole are, in fact, mother and daughter as opposed to two mother/daughter pairs. More importantly, the suggested finding in the paper is that Cep57 works directly with Pericentrin to ensure proper engagement until late mitosis. This model is not supported by the data. At this point, my 4 major concerns listed below preclude recommending this manuscript for publication.

Major concern

1. Measurements of the position of Cep57 and Cep192 should be repeated using SIM. Line-scans should be used to measure the outer diameter of each protein following a fit to a double gaussian curve. This will better represent the data and is necessary to compare to the common measurement found in centrosome literature. Additionally, detailed measurement methods should be described in the Methods section.

2. While it is likely that the two centrioles seen at the poles are the mother and daughter centrioles, the authors should stain mitotic cep57 mutant cells for a mother specific (appendage) protein to ensure these are not two mothers from a previous division error.

3. I am not clear exactly how the authors dismiss an effect on SAC. If there is no change to the NEBD-anaphase timing and there are chromosome segregation errors, then it must be the case that loss of Cep57 suppresses SAC activation. Can the authors explain how this might occur? In the Mad1 experiment, was the level of Mad1 carefully measured under the Cep57 + nocodazole treatment? If a reduction is present, then SAC might not be properly activated. In addition, while the authors show average time of NEBD to anaphase is the same in control and knockdown cells, the distributions of values they obtain is different, with more cells with shorter times in the knockdown condition (Figure 3C). In Figure S3C, 2 of the 6 kinetochores shown do NOT have Mad1 on them, suggesting that the checkpoint is not active on these kinetochores. Finally, the authors say that Cep57 rarely localizes to kinetochores (line 148). The frequency at which this happens and the number of observations should be presented.

4. The presented model of disengagement is not supported. The authors present a model where Cep57 recruits Pericentrin to the centriole to ensure properly engaged centrioles. The data presented is not sufficient to support this mechanism. The authors would need to show the following at minimum to make this conclusion.

- Show that loss of Cep57 leads to loss of PACT. This will establish that Cep57 is the in vivo anchor for Pericentrin through its PACT domain.

- Rescue the disengagement phenotype seen in Pericentrin loss with a Pericentrin chimera where the PACT domain is replaced with Cep57 within the context of full length Pericentrin.

- Show staining for Cep192 and Pericentrin in a Cep57 depleted cell and compare with RNAi to Pericentrin, Pericentrin delta-PACT, and K3154. Images from this manuscript do not indicate that Pericentrin localization is altered in Cep57 mutant as suggested. Depletion of cep57 causes cep192 to extend over 2um wide as shown in figure 2a. But the effect of cep57 on Pericentrin is much less as indicated by Figure 5f (maybe 300 nm change). This is also much different than the effect of K3154 and delta-PACT seen in figure 5d.

Other Concerns

1. The authors must better credit the previous work on Cep57 and its role in pole focusing. Although the two papers (Ref 19 and 20) are mentioned, the authors do not explicitly state that previous work has already shown an increase in multipolar spindles and unfocussed poles. They should state this explicitly in the introduction and again when discussing their own results, such as on line 123.
2. The use of Crenolanib must be justified and specifically explained, as it is not a commonly used inhibitor. Additionally, details of this treatment must be discussed in Methods section.
3. When discussing the measured percentage of multipolar spindles and chromosome segregation phenotypes, it would be important to address the discrepancy between the ~0.8% multipolar spindle phenotype and the 12.6% abnormal chromosome segregation phenotype. How does 0.8% lead to 12.6%? Are the authors suggesting that the tripolar spindles are transient, so called intermediates?
4. Line 193 and in corresponding legend – it is not clear what is meant by “target of the PACT domain”. Are the authors suggesting that Cep57 is the protein that anchors Pericentrin to the centriole via the PACT domain? If so, then this was not tested by the authors. To claim this, cells must be depleted of both Pericentrin and Cep57, while expressing a PACT-GFP construct. Loss of PACT at the centriole will strongly support Cep57 as the main anchor.
5. The authors should explicitly state that the K3154del might also disrupt other protein-protein interaction, especially that K3154 is present in one of the two Calmodulin binding domains of PACT.
6. Line 219 that authors state that PCM is disorganized, but figure S4a shows very organized PCM on each centriole.
7. To properly measure the height of Pericentrin in figure 5f, the authors should include both a distal and proximal centriole marker, or a pan-centriolar marker, to ensure proper orientation and measurement accuracy. It might be the case the entire centriole is shorter. In the control, for example, Pericentrin looks like a ring, suggesting the authors were measuring Pericentrin diameter, not length.
8. Mention of data from reference 28 should be removed as it is unpublished as far as I can tell.

Minor

Line 94 – it is not clear what is meant by “at a constant rate”

Line 96 - The authors should consider better discussing the huge evolutionary gap presented in Figure 1b. They show 5 vertebrates and one single celled eukaryote (a euglenoid). Is this protein present in any organisms between these? How might have Cep57 end up only in such evolutionarily separated species.

Line 136 – 139 – This is an assumption not supported by any data and should be removed or made clear that it is speculation.

Line 144 – given the two difference systems, I suggest replacing “contrary to the previous report” with “in contrast to the role of Cep57 in *Xenopus*” or something similar.

Line 150 – remove “centriolar” from this sentence. Formally, the authors have shown that they can rescue by expressing Cep57, and this Cep57 can localize to the centriole, but they have not excluded the possibility that the rescue arises from its expression elsewhere in the cell.

Line 178 – “implying” should be changed to “suggesting”

Line 181-182 – while the authors show that PACT is necessary, they do not present data that it is sufficient for interaction with Cep57. Thus, the sentence stating that 3139-3216 was capable of binding Cep57 should be changed to 3139-3336. The authors should also mention that they found that AA 1-1962 of Pericentrin, which does NOT contain PACT, also interacts with Cep57 (Figure 4d).

Line 186-188 – similar to the above, the PACT domain per se was not tested for interaction with AKAP. Please change to the PACT-containing fragment of AKAP.

Line 226 – It would be best to remove the words “exact function” as the authors have not proven an exact mechanism for engagement or PCM organization.

Reviewer #3 (Remarks to the Author):

The conserved Cep57-pericentrin module organizes PCM expansion and centriole engagement

Centrosomes are essential organelles that are duplicated once per cell cycle just like DNA. Centrosomes undergo a complex maturation process that ensures functioning of the daughter centrosome as microtubule organizing centre in the next cell cycle. Transition through mitosis is one of the key steps that is needed to decorate the daughter centriole from the previous S phase with pericentriolar material (PCM) such as pericentrin. Relatively little is now about this recruitment process.

Data on the function of CEP57 are heterogeneous and controversial. A 2007 publication in Cell suggests that CEP57 is a kinetochore component. Zhou et al published in Nature Communication that CE57 is a Mis12-interacting kinetochore protein involved in kinetochore targeting of Mad1-Mad2. He et al reported in JBC that CEP57 has a function in cytokinesis. Wu, Q. et al. reported that Cep57, a NEDD1-binding pericentriolar material component, is essential for spindle pole integrity. Interestingly, mutations in CEP57 cause mosaic variegated aneuploidy syndrome.

Watanabe et al analyse in this manuscript the function of CEP57. First, the authors identified a short 36 amino acid sequence that is conserved in the N-terminal coiled coil region of CEP57 family proteins. The authors next analyse CEP57 localizations at centrioles by STED microscopy and show correlation between CEP57 and pericentrin recruitment to the new mother centriole in G1. It seems that a region corresponding to the conserved part of CEP57 (PINC) is essential for centriole targeting. siRNA depletion of CEP57 caused PCM disorganization in mitosis with precociously-disengaged daughter centrioles but surprisingly the PCM of interphase cells was intact. This latter phenotype is inconsistent with the final model. Consistent with the precociously-disengaged daughter centrioles, HeLa cells with CEP57 depletion showed chromosomal segregation errors. They also address the question whether CEP57 has a role in the spindle assembly checkpoint (SAC). In contrast to a previous report in Nature Communication, CEP57 depleted cells have the same mitotic duration as control cells. The authors then switch to MVA1 patient cells with biallelic reduction-of-function CEP57 mutations. MVA patient cells also showed precocious centriole disengagement in mitosis. In Figure 4 the authors test an interaction of CEP57 with pericentrin. Using overexpression conditions, they show co-immunoprecipitation of pericentrin and CEP57. Their data suggest that PINC region and PACT of pericentrin are important for this interaction. While the co-IP experiment suggests that CEP57 and pericentrin are present in common complexes, the in vitro binding data of Figure 4g are less convincing (see specific points). In the final Figure 5 they

show that an amino acid deletion or partial deletion of the in PACT of pericentrin affects CEP57 interaction by IP. These mutations in pericentrin also premature centriole disengagement in mitosis as was seen upon CEP57 depletion.

This manuscript shows that the pericentrin-CEP57 interaction is important for PCM organization in mitosis. The PINC region of CEP57 and the PACT of pericentrin are important for PCM stabilization. However, the lack of interphase PCM organization argues against the simple CEP57-pericentrin recruitment model. Thus, presently it is unclear how CEP57 functions at centrosomes in mitosis. At the end, this manuscript does not substantially go beyond what has been published by He et al. in Cell Research. The conclusion of He et al. was: "Depletion of Cep57 leads to unaligned chromosomes and a multipolar spindle, which is induced by PCM fragmentation." Additional functional experiments on CEP57 and the regulation of CEP57 in mitosis are needed to justify publication in Nature Communication.

Major points.

1. Based on 36 aa homology in a coiled-coil region, I find it risky to conclude that CEP57 is a conserved factor across species.
2. The authors conclude that CEP57 interacts with the PACT domain of PCNT. Therefore, CEP57 should co-localize with the C-terminus of PCNT but not the N-terminus of PCNT. To test this, STED analysis in Figure 1 needs to be performed with N- and C-terminal PCNT antibodies in comparison to CEP57. This should be done in interphase and mitosis. Mitosis is especially important because CEP57 depletion is causing mitotic phenotypes.
3. One of the main conclusions of this manuscript is that CEP57 recruits PCNT to the centriole. This model predicates that centriole binding of PCNT is defective in CEP57 depleted cells (interphase, perhaps mitosis). However, the authors state on lines 100-101 that "Cep57 did not lead to any defects in PCM formation and centriole duplication in interphase cells." This finding is inconsistent with the model.
4. In Figure 4 the authors show that CEP57 interacts with the PACT of PCNT. However these data have problems. First, binding efficiency is low (Fig. 4g). Second, binding is done with recombinant PACT without calmodulin binding. However, calmodulin is an essential co-factor of PACT. Third, HA-CEP57 in the Coomassie blue stained gel is a solid band. This amount should give a very strong signal in the immunoblot. However, in the immunoblot above the CEP57 signal in lane 1 is as in lane 5. CEP57 in lane 5 is not visible in the Coomassie Blue stained gel. Thus, the loading of the samples must be unequal.
5. Figure 3a: I cannot see the lagging chromosomes – resolution of the chromosomes is not good enough.
6. Do MVA patient cells also have lagging chromosomes?
7. In Figure 4c there is no FLAG signal in the input.

Comments on points raised by the reviewers

We thank all the reviewers of our original manuscript for their critical reading and for their useful and constructive comments (typed in blue), which we fully addressed in the revised version with new data (answers typed in black). Following their suggestions, we substantially altered the manuscript, as detailed below.

Reviewers' comments:

Reviewer #1 (Remarks to the Author):

In this manuscript, Kitagawa and his colleagues reported roles of Cep57 in the centrosome. They revealed that Cep57 is located at the proximal end of mother centriole. Depletion of Cep57 causes PCM disorganization and precocious centriole disengagement during mitosis. As a result, the disengaged daughter centrioles function as spindle poles, leading to chromosome mis-segregation and aneuploidy. Similar defects were observed in mosaic variegated aneuploidy syndrome patient cells with cep57 mutations. They also observed a specific interaction of Cep57 with the PACT domain of pericentrin. Based on the results, the authors proposed that the Cep57-pericentrin complex provides a critical interphase between centriole core and PCM, which is crucial for the higher-order structure of the PCM.

This work revealed how Cep57 works at the centrosome. These results are novel and potentially important for elucidating structural and functional linkage between centrioles and PCM. Overall, their experiments seemed to be carefully carried out and the results looked trustful. This work is worth to be published in Nature Communication. The follows are major points to be resolved prior to publication.

> We thank this reviewer for the supportive comment on our manuscript.

1. The authors may have to explain why their results are different from the previous papers in which Cep57 functions in the kinetochore. One of critical differences between this work and Zhou et al. (2016) may be the immunostaining of Cep57 in mitotic cells. The Cep57 signals were coimmunostained with CREST, CENP-A and Mis12 in Zhou et al. (2016). On the other hand, this manuscript did not show an immunostaining image of the whole mitotic cells with the Cep57 antibody.

> This reviewer raised a critical question on the discrepancy of Cep57 localization at kinetochores. We followed the same protocol by which Zhou et al. detected Cep57 localization at kinetochores, except for using two different antibodies against Cep57. Although Cep57 signals were efficiently detected at centrosomes, we could not find any trace of Cep57 signal at kinetochores. We then tested different fixation methods by one of which we managed to detect Cep57 signals at kinetochores even though the signal intensity was weak (new data added in Figure S3d of the revised manuscript). We also performed fluorescence live cell imaging with Cep57-mCherry, but we hardly detected Cep57 signals at kinetochores (data not shown). We therefore conclude that Cep57 could modestly localize to kinetochores in human cells. However, contrary to the Cep57 knockdown phenotype that was reported in Zhou et al. (2016), we could not observe significant defects in the spindle assembly

checkpoint and mitotic duration (new data in Figure 3c and Supplementary Figure 3e,f). We therefore assume the specific function of Cep57 at kinetochores is likely very limited in the context of chromosome segregation. We are therefore sure that the PCM disorganization and precocious centriole disengagement provoked by Cep57 depletion should be irrelevant to potential function of Cep57 at kinetochores.

2. To enforce their proposal, knockdown-rescue experiments of Fig. 2f,g may be performed with HA-Cep57 deletion ($\Delta 68-103$) and truncated mutants (1-530; Zhou et al., 2016).

>We thank this reviewer for suggesting this experiment. Following this suggestion, we conducted the rescue experiment with HA-Cep57 deletion mutant lacking the PINC motif. As indicated in the original manuscript, the Cep57 mutant failed to localize to centrioles. Consistently, we demonstrated that expression of the Cep57 mutant did not rescue the phenotype of siCep57 (new data in Supplementary Figure 5e, mentioned in the revised manuscript (page 13, line 266-271)). Regarding the truncated mutant (1-530) that this reviewer mentioned, we suspect this might refer to the Mad1 mutant (Zhou et al. 2016) since Cep57 encompasses 500 a.a.

3. The authors reported that depletion of Cep57 did not lead to any defects in PCM formation and centriole duplication in interphase cells. It is worth to determine that depletion of Cep57 hardly affect daughter centriole recruitment of Cep152 in interphase cells.

>Prompted by this reviewer' comment, we tested the loading of Cep152 at centrioles upon Cep57 depletion, and found that it was not affected in interphase human cells (new data in Supplementary Figure 2a).

4. Precocious centriole disengagement was observed in Cep57-depleted cells (Figs. 3a). The separated centrioles with ectopic MTOC activities eventually became clustered to form pseudo-bipoles (Fig. 2h, Supplementary Fig. 2i,j). Nonetheless, the authors did not observe a delay in NEBD to anaphase onset time in Cep57-depleted cells (Fig. 3c). In fact, Zhou et al. (2016) reported a shortening of the NEBD to anaphase onset time in Cep57-depleted cells. The authors need to explain this observation, based on your proposal.

>This reviewer is correct in stating that mitotic delay could happen when the precociously separated centrioles acquire ectopic MTOC activities during mitosis. We carefully reanalyzed the live cell imaging movies with Cep57-depleted cells, and confirmed a significant mitotic delay when the cells exhibited chromosome misalignment because of ectopic MTOCs formation, including multipolar spindle formation (new quantification data in Supplementary Figure 3d). It seems that the precocious disengagement of centrioles itself was not sufficient to induce mitotic arrest, since these centrioles were clustered into pseudo-bipoles in a relatively quick manner. As this reviewer mentioned, Zhou et al. (2016) reported a shortening of the NEBD to anaphase onset time in Cep57-depleted cells. However, we could not observe such a significant shortening of mitosis in Cep57-depleted cells in our condition (Figure 3c). To address this further, we repeated the same protocol, as Zhou et al. (2016) performed, to measure the NEBD to anaphase onset time under nocodazole-treatment, however, we could not observe a

decrease in the mitotic duration, compared to the control, whereas the Mad1-depletion bypassed the SAC (spindle assembly checkpoint) pathway and so efficiently shortened the NEBD to anaphase onset time (new data in Supplementary Figure 3d). We therefore conclude that the function of Cep57 for the regulation of SAC seems to be limited and not that critical, at least, in our condition. This conclusion is more explicitly stated in the revised manuscript (page 8 and 9).

5. The model proposes that pericentrin interacts with Cep57 at the mother centriole. This configuration may be acceptable only in interphase cells. If the model is correct, then precocious centriole disengagement should be observed in interphase cells. According to Lawo et al. (2012), the PCM structure of the mitotic cells are quite different with massive accumulation of pericentrin and other components. Therefore, the model should be corrected for mitotic cells.

>We thank this reviewer for the valuable comment on our model and we understand the argument. We apologize for not having sufficiently explained the model, but we conclude that the Cep57-pericentrin interaction is crucial for the expansion and rearrangement of PCM and timely centriole disengagement during mitosis. In interphase, although both Cep57 and Pericentrin colocalize at the vicinity of centrioles, depletion of Cep57 or Pericentrin did not affect the organization of PCM and centriole engagement. We therefore assume that a redundant mechanism somehow maintains centriole engagement in interphase, independently of the Cep57-Pericentrin interaction. Regarding mitotic processes, our model is that overall rearrangement of PCM components takes place for coupling PCM expansion and centriole engagement rather than just recruiting additional PCM components on top of the pre-existing PCM layer. We found that specific interaction of Cep57 and Pericentrin is critical for this process and proper centriole engagement until late mitosis. Prompted by this reviewer's comment, we now corrected the model in Figure 6 so as to fit our claim.

6. The authors need to define how to discriminate precocious centriole disengagement.

>We thank this reviewer for raising this issue, which we defined in the revised manuscript (page 5, line 109-111). Basically, we discriminate precocious centriole disengagement in Cep57-depleted cells, when the distance between mother and daughter centrioles, in prometaphase or metaphase, is at least twice more than that of the control or interphase cells.

Reviewer #2 (Remarks to the Author):

The manuscript by Watanabe and colleagues explores the role of Cep57 in centriole engagement. They show that upon Cep57 depletion the mother-daughter centriole prematurely disengage as early as prophase, when centrioles normally disengage in late mitosis. The authors show that early disengagement leads to disorganized PCM, splayed spindle poles, and ultimately defects in chromosome segregation. Given that Pericentrin also

shows a similar centriole disengagement and PCM phenotypes upon depletion, the authors explore the possibility that Cep57 and Pericentrin function together. They indeed show direct binding of these two proteins that depends on the newly identified and conserved PINC domain in Cep57 and the PACT domain in Pericentrin. The authors also suggest that the disease linked mutation K3154del in Pericentrin might cause the disease as a result of loss of interaction with Cep57.

This study is potentially interesting as it more clearly explains the previously shown multipolar spindle phenotype by identifying a centriole disengagement phenotype. As you will see below, the authors must do additional experiments to ensure that the two distant centrioles at the pole are, in fact, mother and daughter as opposed to two mother/daughter pairs. More importantly, the suggested finding in the paper is that Cep57 works directly with Pericentrin to ensure proper engagement until late mitosis. This model is not supported by the data. At this point, my 4 major concerns listed below preclude recommending this manuscript for publication.

>We appreciate this reviewer for supportive and constructive comments. As detailed below, we addressed his/her major and minor concerns in full.

Major concern

1. Measurements of the position of Cep57 and Cep192 should be repeated using SIM. Line-scans should be used to measure the outer diameter of each protein following a fit to a double gaussian curve. This will better represent the data and is necessary to compare to the common measurement found in centrosome literature. Additionally, detailed measurement methods should be described in the Methods section.

>We thank this reviewer for valuable suggestion on the detailed position of Cep57 within centrioles. Prompted by this comment, we performed STED microscopy analysis to precisely measure the outer diameter of each protein. The ring radius of Cep57 and Cep192 were measured using the ImageJ plugin "Radial Profile Extended". The obtained profile was then fitted with a Gaussian curve and the distance between the center of the ring and the peak of the Gaussian curve was defined as the radius. The diameter of the ring was defined as twice the radius. We again found that the diameter of Cep57 was close to that of Cep192. This method is more appropriate and is now explicitly described in the Methods section (page 37).

2. While it is likely that the two centrioles seen at the poles are the mother and daughter centrioles, the authors should stain mitotic cep57 mutant cells for a mother specific (appendage) protein to ensure these are not two mothers from a previous division error.

>As this reviewer mentioned, it would be important to make sure a pair of centrioles at the poles are indeed mother and its daughter centrioles. We therefore tested ODF2 staining which marks mother centrioles and found a single ODF2-positive focus out of two separate centrioles at a spindle pole in Cep57-depleted cells. This observation indicates that they are a pair of centrioles and supports the phenotype of precocious centriole disengagement. This new data is now added in Figure S2d and mentioned in the revised manuscript (page 5-6, line113-117).

3. I am not clear exactly how the authors dismiss an effect on SAC. If there is no change to the NEBD-anaphase timing and there are chromosome segregation errors, then it must be the case that loss of Cep57 suppresses SAC activation. Can the authors explain how this might occur? In the Mad1 experiment, was the level of Mad1 carefully measured under the Cep57 + nocodazole treatment? If a reduction is present, then SAC might not be properly activated. In addition, while the authors show average time of NEBD to anaphase is the same in control and knockdown cells, the distributions of values they obtain is different, with more cells with shorter times in the knockdown condition (Figure 3C). In Figure S3C, 2 of the 6 kinetochores shown do NOT have Mad1 on them, suggesting that the checkpoint is not active on these kinetochores. Finally, the authors say that Cep57 rarely localizes to kinetochores (line 148). The frequency at which this happens and the number of observations should be presented.

>This reviewer is correct in arguing that SAC activation and resulting mitotic delay should be induced when the separated centrioles with ectopic MTOC activity cause chromosome segregation error in Cep57-depleted cells. We carefully reanalyzed the live cell imaging movies of Cep57-depleted cells, and confirmed a significant mitotic delay when the cells exhibited apparent chromosome segregation errors because of ectopic MTOCs formation, including chromosome misalignment, lagging chromosome and multipolar spindle formation (new quantification data in Figure S3e). This observation indicates that SAC is still active in the absence of Cep57. On the other hand, it seems that the precocious disengagement of centrioles itself was not sufficient to induce mitotic arrest, since these centrioles were clustered into pseudo-bipoles in a relatively quick manner. Therefore, as a whole, there was not significant increase in mitotic duration upon Cep57 depletion.

>Following the question of this reviewer, we quantified the Mad1 levels at kinetochores under nocodazole-treatment in Cep57-depleted cells. On the contrary to what Zhou et al. reported, the Mad1 signal intensity was not reduced and rather a bit increased, compared to the control (~1.1 fold, N>1000). So, as this reviewer pointed out, the original Figure S3c is not appropriate as a representative, we replaced it with the current panels in the revised manuscript (new data in Figure S3g,h).

>As this reviewer mentioned, Zhou et al. (2016) reported a shortening of the NEBD to anaphase onset time in Cep57-depleted cells. Following this reviewer's comment, we re-analyzed distribution of the values, however, we could not observe such a significant shortening of mitosis in Cep57-depleted cells in our condition (no statistical difference, Figure 3c). To address this further, we repeated the same protocol, as Zhou et al. (2016) performed, to measure the NEBD to anaphase onset time under nocodazole-treatment, however, we could not observe a decrease in the mitotic duration, compared to the control, whereas the Mad1-depletion bypassed the SAC pathway and so efficiently shortened the NEBD to anaphase onset time (new data in Figure S3f). We therefore conclude that the function of Cep57 for the regulation of SAC seems to be limited and not that critical, at least, in our condition. This conclusion is more explicitly stated in the revised manuscript (page 8 and 9).

> This reviewer raised a critical question on the discrepancy of Cep57

localization at kinetochores. We followed the same protocol by which Zhou et al. detected Cep57 localization at kinetochores, except for using two different antibodies against Cep57. Although Cep57 signals were efficiently detected at centrosomes, we could not find any trace of Cep57 signal at kinetochores. We then tested different fixation methods by one of which we managed to detect Cep57 signals at kinetochores (new data added in Figure S3d of the revised manuscript, this information is now explicitly described in page 8 and 27). We also performed fluorescence live cell imaging with Cep57-mCherry, but we hardly detected Cep57 signals at kinetochore (data not shown). We therefore conclude that Cep57 could modestly localize to kinetochores in human cells. However, contrary to the Cep57 knockdown phenotype that was reported in Zhou et al. (2016), we could not observe significant defects in the spindle assembly checkpoint during mitosis (new data in Figure 3c and S3f). We therefore assume the specific function of Cep57 at kinetochores is likely very limited in the context of chromosome segregation. We therefore think that the PCM disorganization and precocious centriole disengagement provoked by Cep57 depletion should be irrelevant to potential function of Cep57 at kinetochores.

4. The presented model of disengagement is not supported. The authors present a model where Cep57 recruits Pericentrin to the centriole to ensure properly engaged centrioles. The data presented is not sufficient to support this mechanism. The authors would need to show the following at minimum to make this conclusion.

>We thank this reviewer for constructive suggestions on the following experiments.

- Show that loss of Cep57 leads to loss of PACT. This will establish that Cep57 is the *in vivo* anchor for Pericentrin through its PACT domain.

>We conducted this experiment and found that Cep57 depletion impaired centriolar loading of PACT, indicating that Cep57 is indeed an anchor for the PACT domain (new data in Figure 3h,i, mentioned in the revised manuscript (page 11, line 228-236)).

- Rescue the disengagement phenotype seen in Pericentrin loss with a Pericentrin chimera where the PACT domain is replaced with Cep57 within the context of full length Pericentrin.

>As this reviewer suggested this excellent experiment, we designed a construct of the Pericentrin chimera in which the PACT was replaced with Cep57. As expected, this mutant efficiently rescued the disengagement phenotype of Pericentrin knockdown, indicating that the Cep57-PACT interaction is sufficient for centrosomal localization and the function of Pericentrin (new data in Figure 5f,g,h, and now mentioned in page 13, line 272-277).

- Show staining for Cep192 and Pericentrin in a Cep57 depleted cell and compare with RNAi to Pericentrin, Pericentrin delta-PACT, and K3154. Images from this manuscript do not indicate that Pericentrin localization is altered in Cep57 mutant as suggested. Depletion of cep57 causes cep192 to extend over 2um wide as shown in figure 2a. But the effect of cep57 on Pericentrin is much less as indicated by Figure 5f (maybe 300 nm change). This is also much different than the effect of K3154 and delta-PACT seen in figure 5d.

> Prompted by the comment from this reviewer, we reassessed the effect of Cep57 depletion on Pericentrin and Cep192 localization at centrosomes. We found that the patterns of Pericentrin and Cep192 at centrosomes were similarly disorganized and spread out upon Cep57 depletion (new data in Figure S5b). The extent of PCM dispersion was comparable to Pericentrin RNAi and expression of the Pericentrin mutants.

Other Concerns

1. The authors must better credit the previous work on Cep57 and its role in pole focusing. Although the two papers (Ref 19 and 20) are mentioned, the authors do not explicitly state that previous work has already shown an increase in multipolar spindles and unfocused poles. They should state this explicitly in the introduction and again when discussing their own results, such as on line 123.

> We thank this reviewer for pointing this out. We now mention the previous finding on the role of Cep57 in pole focusing in the introduction of the revised manuscript (page 3). In addition, we mention their results more explicitly in the discussion (page 14-15).

2. The use of Crenolanib must be justified and specifically explained, as it is not a commonly used inhibitor. Additionally, details of this treatment must be discussed in Methods section.

> We specified and explained the effect of Crenolanib on pole focusing in the revised manuscript (page 7, line 140-141). The treatment condition that we used in this study was also described in the method section and figure legends (page 36 and 26).

3. When discussing the measured percentage of multipolar spindles and chromosome segregation phenotypes, it would be important to address the discrepancy between the ~0.8% multipolar spindle phenotype and the 12.6% abnormal chromosome segregation phenotype. How does 0.8% lead to 12.6%? Are the authors suggesting that the tripolar spindles are transient, so called intermediates?

> We thank this reviewer for raising this issue. Regarding this, we have to apologize for having made a wording mistake in the sentence and for the confusion. The correct sentence in here is 'Live-cell imaging showed that Cep57-depleted cells exhibited chromosomal segregation errors (12.6%, compared to 2.6% in control cells) SUCH AS (not 'because of' in the original manuscript) chromosome misalignment, chromosome lagging and tripolar spindle formation, and also multi-nuclei formation' We now corrected the sentence in the revised manuscript (page 7, line 149-152) We assume that a pseudo-bipolar spindle with separate centriole pairs can induce chromosome missegregation. In other words, formation of multiple MTOCs could affect the fidelity of chromosome segregation even though they are clustered into two spindle poles. This could explain the difference between the ~0.8% multipolar spindle phenotype and the 12.6% abnormal chromosome segregation phenotype.

4. Line 193 and in corresponding legend – it is not clear what is meant by "target of the PACT domain". Are the authors suggesting that Cep57 is the

protein that anchors Pericentrin to the centriole via the PACT domain? If so, then this was not tested by the authors. To claim this, cells must be depleted of both Pericentrin and Cep57, while expressing a PACT-GFP construct. Loss of PACT at the centriole will strongly support Cep57 as the main anchor.

>As mentioned above, we conducted this experiment and found that Cep57 depletion abolished the centriolar recruitment of PACT-GFP. This observation supports the claim that Cep57 is the protein that mainly anchors Pericentrin to the centriole via the PACT domain. On the other hand, we observed that Pericentrin full-length still localized to centrioles even upon Cep57 depletion, suggesting that another anchor protein should load Pericentrin to centrioles through binding to other region (but not the PACT) of Pericentrin. However, the Cep57-PACT of Pericentrin interaction is absolutely needed for proper arrangement of Pericentrin and PCM expansion during mitosis as the dispersion of Pericentrin in mitosis was observed upon Cep57 depletion. We added the new data in Figure S5b and mentioned this result in the revised manuscript (page 12, lane 263-264).

5. The authors should explicitly state that the K3154del might also disrupt other protein-protein interaction, especially that K3154 is present in one of the two Calmodulin binding domains of PACT.

>We appreciate the comment from this reviewer. It is known that two Calmodulin binding domains exist in the DPLP PACT domain in *Drosophila*, but in human, there seems to be only one Calmodulin binding domain in Pericentrin (c.f. Calmodulin Target Database). As far as we checked, K3154 does not reside within the Calmodulin binding domain of PACT of human Pericentrin (spanning 3193-3212 a.a.). However, we agree with this reviewer that K3154del might also disrupt other critical protein-protein interaction, so we modified this statement in the revised manuscript (page 11, lane 242-244).

6. Line 219 that authors state that PCM is disorganized, but figure S4a shows very organized PCM on each centriole.

>We apologize for the confusion and not having more clearly explained this phenotype. We observed PCM disorganization in the case of expression of the Pericentrin PCAT mutants, but not Pericentrin knockdown solely (Figure 5d and S4a). Our interpretation is that Pericentrin depletion, but not expression of the PACT mutants, leads to loss of PCM expansion and this defect prevents the occurrence of PCM disorganization. It is also known that phosphorylation of pericentrin by Plk1 is needed for PCM expansion (Lee et al. 2011). This issue is now discussed in the revised manuscript (page 14). As this reviewer mentioned, in Figure S4a, PCM around centrioles is small, but organized upon Pericentrin depletion. In this situation, precocious centriole disengagement could happen, but not PCM disorganization because of loss of PCM expansion.

7. To properly measure the height of Pericentrin in figure 5f, the authors should include both a distal and proximal centriole marker, or a pan-centriolar marker, to ensure proper orientation and measurement accuracy. It might be the case the entire centriole is shorter. In the control, for example, Pericentrin looks like a ring, suggesting the authors were measuring Pericentrin diameter, not length.

>We appreciate this experimental suggestion. Accordingly, we tried triple-staining to mark distal and proximal ends of centrioles to ensure accurate orientation of centrioles. Unfortunately, we found that the far-red labeling of Cep152 antibodies to mark the proximal end gave us a fuzzy signal which was not appropriate for measuring the centriole length accurately. We then tested the staining with Cep152 and ODF2 antibodies to concurrently mark proximal and distal ends in the same color (red), however, it did not work as well. In the original experiment, we collected side views of centrioles by judging from the bar-shaped pattern of ODF2 (Tateishi et al. (2013) JCB, Fig. 4A; Ibi et al. (2011) JCS, Fig. 2B) and measured the height of Pericentrin in a vertical direction. Pericentrin should look like a ring only when we observe it from the top view and, in such case, ODF2 should also look like a ring. As far as we tested, this strategy worked the best for us to measure the Pericentrin height.

8. Mention of data from reference 28 should be removed as it is unpublished as far as I can tell.

> Thanks for the comment. Because we find that it was already published, we leave it as it is.

Minor

Line 94 – it is not clear what is meant by “at a constant rate”

>We replace “at a constant rate” with “occasionally”.

Line 96 - The authors should consider better discussing the huge evolutionary gap presented in Figure 1b. They show 5 vertebrates and one single celled eukaryote (a euglenoid). Is this protein present in any organisms between these? How might have Cep57 end up only in such evolutionarily separated species.

>We thank this reviewer for raising this important issue. As far as we looked into the existence of Cep57 among species, we found Cep57 homologues in most vertebrates, but apart from them, we could find it only in Trypanosoma and yeasts so far. Unfortunately, our blast search did not hit any other invertebrate organisms presumably due to the short consensus sequence. However, we speculate that its homologues may be present in other invertebrates. For instance, given that the PACT domains of DPLP (*Drosophila*) and Pericentrin (human) share homology, we speculate that a homologue of Cep57, the binding partner of Pericentrin, could exist in *Drosophila* as well. Accordingly, we now added more discussion, and mentioned that this protein is well conserved in vertebrates and also found in Trypanosoma and yeasts (page 4 and 15).

Line 136 – 139 – This is an assumption not supported by any data and should be removed or made clear that it is speculation.

>This reviewer is correct in stating that this claim seems an assumption. We now made it clear that this is an assumption by modifying these sentences. However, this claim is based on our observations as shown in Figure S2j and S3a. We indicated the figure number with this claim in the revised manuscript (page 7 and 8, lane 153-158).

Line 144 – given the two difference systems, I suggest replacing “contrary to the previous report” with “in contrast to the role of Cep57 in Xenopus” or something similar.

>The previous work which showed the role of kinetochore Cep57 in SAC activation was done in human cells (Zhou et al. (2016)), and we performed similar experiments using human cell lines. So we leave the phrase as it is.

Line 150 – remove “centriolar” from this sentence. Formally, the authors have shown that they can rescue by expressing Cep57, and this Cep57 can localize to the centriole, but they have not excluded the possibility that the rescue arises from its expression elsewhere in the cell.

>We thank this reviewer for the suggestion. We agree with this opinion and removed ‘centriolar’ from this sentence.

Line 178 – “implying” should be changed to “suggesting”

>We did, thank you very much.

Line 181-182 – while the authors show that PACT is necessary, they do not present data that it is sufficient for interaction with Cep57. Thus, the sentence stating that 3139-3216 was capable of binding Cep57 should be changed to 3139-3336. The authors should also mention that they found that AA 1-1962 of Pericentrin, which does NOT contain PACT, also interacts with Cep57 (Figure 4d).

> We have changed the sentence to “short fragments of PCNT (aa 3139-3336 and aa 2945-3216) were capable of binding to Cep57” (page 10, line 205-206). Following the comment from this reviewer, we mentioned 1-1962 aa of Pericentrin also interacts with Cep57 in the revised manuscript (page 21, line 433-435).

Line 186-188 – similar to the above, the PACT domain per se was not tested for interaction with AKAP. Please change to the PACT-containing fragment of AKAP.

>Similarly, we changed it to the PACT-containing fragment of AKAP in the corresponding statement (page 10, line 210-212).

Line 226 – It would be best to remove the words “exact function” as the authors have not proven an exact mechanism for engagement or PCM organization.

>We followed this suggestion because we agree with the comment from this reviewer. Using “exact function” would not be appropriate here (page 13, line 278).

Reviewer #3 (Remarks to the Author):

The conserved Cep57-pericentrin module organizes PCM expansion and centriole engagement

Centrosomes are essential organelles that are duplicated once per cell cycle just like DNA. Centrosomes undergo a complex maturation process that ensures functioning of the daughter centrosome as microtubule organizing centre in the next cell cycle. Transition through mitosis is one of the key steps that is needed to decorate the daughter centriole from the previous S phase with pericentriolar material (PCM) such as pericentrin. Relatively little is now about this recruitment process.

Data on the function of CEP57 are heterogeneous and controversial. A 2007 publication in *Cell* suggests that CEP57 is a kinetochore component. Zhou et al published in *Nature Communication* that CEP57 is a Mis12-interacting kinetochore protein involved in kinetochore targeting of Mad1-Mad2. He et al reported in *JBC* that CEP57 has a function in cytokinesis. Wu, Q. et al. reported that Cep57, a NEDD1-binding pericentriolar material component, is essential for spindle pole integrity. Interestingly, mutations in CEP57 cause mosaic variegated aneuploidy syndrome.

Watanabe et al analyse in this manuscript the function of CEP57. First, the authors identified a short 36 amino acid sequence that is conserved in the N-terminal coiled coil region of CEP57 family proteins. The authors next analyse CEP57 localizations at centrioles by STED microscopy and show correlation between CEP57 and pericentrin recruitment to the new mother centriole in G1. It seems that a region corresponding to the conserved part of CEP57 (PINC) is essential for centriole targeting. siRNA depletion of CEP57 caused PCM disorganization in mitosis with precociously-disengaged daughter centrioles but surprisingly the PCM of interphase cells was intact. This latter phenotype is inconsistent with the final model. Consistent with the precociously-disengaged daughter centrioles, HeLa cells with CEP57 depletion showed chromosomal segregation errors. They also address the question whether CEP57 has a role in the spindle assembly checkpoint (SAC). In contrast to a previous report in *Nature Communication*, CEP57 depleted cells have the same mitotic duration as control cells. The authors then switch to MVA1 patient cells with biallelic reduction-of-function CEP57 mutations. MVA patient cells also showed precocious centriole disengagement in mitosis. In Figure 4 the authors test an interaction of CEP57 with pericentrin. Using overexpression conditions, they show co-immunoprecipitation of pericentrin and CEP57. Their data suggest that PINC region and PACT of pericentrin are important for this interaction. While the co-IP experiment suggests that CEP57 and pericentrin are present in common complexes, the in vitro binding data of Figure 4g are less convincing (see specific points). In the final Figure 5 they show that an amino acid deletion or partial deletion of the in PACT of pericentrin affects CEP57 interaction by IP. These mutations in pericentrin also premature centriole disengagement in mitosis as was seen upon CEP57 depletion.

This manuscript shows that the pericentrin-CEP57 interaction is important for PCM organization in mitosis. The PINC region of CEP57 and the PACT of

pericentrin are important for PCM stabilization. However, the lack of interphase PCM organization argues against the simple CEP57-pericentrin recruitment model. Thus, presently it is unclear how CEP57 functions at centrosomes in mitosis. At the end, this manuscript does not substantially go beyond what has been published by He et al. in Cell Research. The conclusion of He et al. was: "Depletion of Cep57 leads to unaligned chromosomes and a multipolar spindle, which is induced by PCM fragmentation." Additional functional experiments on CEP57 and the regulation of CEP57 in mitosis are needed to justify publication in Nature Communication.

>We thank this reviewer for the constructive comments on our study. First, although our explanation might not be sufficiently clear in the original manuscript, our claim is that the Cep57-Pericentrin interaction is crucial for proper expansion and organization of mitotic PCM (Figures 2,3, and 5), but is dispensable for centriolar recruitment of Pericentrin and interphase PCM. Pericentrin without PACT domain can still localize to centrosomes, presumably by anchoring another centrosome protein except for Cep57, in interphase. It seems that PCM components are actively recruited to centrosomes and the binding mode of the components and its arrangement in the PCM change to fit proper expansion of functional PCM in mitosis. In this process, we found that the Cep57-Pericentrin interaction is critical because its loss led to PCM disorganization and associated precocious centriole disengagement. A series of mutant analyses that we performed in Figure 4 and 5, clearly demonstrated that the specific interaction between Cep57 and Pericentrin PACT is needed for proper PCM expansion and timely centriole disengagement during mitosis. Moreover, the additional experiments further support the claim as detailed below.

>He et al. in Cell Research roughly described the phenotype of Cep57RNAi. Their observation is that, in Cep57-depleted cells, PCM fragmentation induces multipolar spindle formation. We uncovered a molecular mechanism that the Cep57-Pericentrin complex regulates mitotic PCM organization. Furthermore, although the conclusion of He et al. is that the PCM fragmentation is a cause of multipolar spindle formation in Cep57-depleted cells, we conclude that the precocious centriole disengagement and resulting ectopic MTOC formation is the direct cause of the multipolar spindle formation. In fact, apparent multipolar phenotype is relatively rare, but a pseudo-bipolar spindle with multiple MTOCs is a major phenotype in Cep57-depleted cells, which can induce chromosome segregation error. Thus, our findings provide insights into how PCM expansion and arrangement for timely centriole disengagement are coupled by revealing the function of the Cep57-Pericentrin complex in mitosis. Together with additional experiments that this reviewer requested, we believe that our results illustrate the new function of Cep57 at mitosis, as detailed below.

Major points.

1. Based on 36 aa homology in a coiled-coil region, I find it risky to conclude that CEP57 is a conserved factor across species.

>We thank the comment from this reviewer. As far as we looked into the existence of Cep57 among species, we found Cep57 homologues in most vertebrates (new data in Figure S1a), but apart from them, we could find it

only in Trypanosoma and yeasts so far. Unfortunately, our blast search did not hit any other invertebrate organisms presumably due to the short consensus sequence. However, we speculate that its homologues may be present in other invertebrates. Accordingly, we now changed this statement in a modest manner, and mentioned that this protein is well conserved in vertebrates and also found in Trypanosoma and yeasts (page 4 and 15).

2. The authors conclude that CEP57 interacts with the PACT domain of PCNT. Therefore, CEP57 should co-localize with the C-terminus of PCNT but not the N-terminus of PCNT. To test this, STED analysis in Figure 1 needs to be performed with N- and C-terminal PCNT antibodies in comparison to CEP57. This should be done in interphase and mitosis. Mitosis is especially important because CEP57 depletion is causing mitotic phenotypes.

>We thank this reviewer for the valuable suggestion. Accordingly, we conducted STED analysis to test the co-localization of Cep57 and Pericentrin. As this reviewer suggested, using antibodies against N- and C-terminus of PCNT, we demonstrated that the position of C-terminus of PCNT is much closer to Cep57 at the inner centrosome than that of N-terminus of PCNT. This observation is true for both in interphase and mitosis. This new data is now added in Figure S4g and mentioned in the revised manuscript (page 11, line 231-236).

3. One of the main conclusions of this manuscript is that CEP57 recruits PCNT to the centriole. This model predicates that centriole binding of PCNT is defective in CEP57 depleted cells (interphase, perhaps mitosis). However, the authors state on lines 100-101 that “Cep57 did not lead to any defects in PCM formation and centriole duplication in interphase cells.” This finding is inconsistent with the model.

>We apologize for not having sufficiently explained this statement. Our model is that the Cep57-pericentrin interaction works for the mitotic expansion and rearrangement of PCM for proper centriole engagement until late mitosis. On the other hand, in interphase, we observed that Pericentrin full-length still localized to centrioles even upon Cep57 depletion, suggesting that another anchor protein should load Pericentrin to centrioles through binding to other region (but not the PACT) of Pericentrin. However, the Cep57-PACT of Pericentrin interaction is absolutely needed for proper rearrangement of Pericentrin in mitotic PCM and PCM expansion as the dispersion of Pericentrin in mitosis was observed upon Cep57 depletion. We therefore conclude that the Cep57-pericentrin interaction is not needed for centriole engagement and duplication in interphase cells, but crucial for the timely centriole disengagement in mitotic cells. We accordingly modified the model in Figure 5h so as to fit our claim.

4. In Figure 4 the authors show that CEP57 interacts with the PACT of PCNT. However these data have problems. First, binding efficiency is low (Fig. 4g). Second, binding is done with recombinant PACT without calmodulin binding. However, calmodulin is an essential co-factor of PACT. Third, HA-CEP57 in the Coomassie blue stained gel is a solid band. This amount should give a very strong signal in the immunoblot. However, in the immunoblot above the CEP57 signal in lane 1 is as in lane 5. CEP57 in lane 5 is not visible in the

Coomassie Blue stained gel. Thus, the loading of the samples must be unequal.

>We thank this reviewer for the critical comments on the *in vitro* binding assay with purified proteins of Cep57 and PCNT.

First, prompted by this reviewer's comment, we repeated this experiment in different conditions (e.g. buffer, protein amounts, loading volume, etc). The efficiency is now improved as shown in new Figure 4f. We also found that this interaction was reproducibly and robustly detected by different combination of tagging (GST or MBP) and yeast two-hybrid assay (new data in Figure 4f). Second, we appreciate the comment on Calmodulin as a cofactor of PACT. We tested the effect of Calmodulin on the PCM organization and centriole engagement in human cells, and did not find any significant defects upon Calmodulin depletion by two different siRNAs (new data in Figure S4h,i). Furthermore, although two Calmodulin binding domains exist in the DPLP PACT domain in *Drosophila*, there seems to be only one Calmodulin binding domain in human PCNT (c.f. Calmodulin Target Database). GFP-PACT Δ CaM that does not bind to Calmodulin was still able to localize to centrosomes in human cells (Gillingham et al. (2000) EMBO, Fig 3B). The PCNTK3154del mutant, the mutated residue of which resides outside of the Calmodulin binding motif (3193-3212 aa), is defective in interacting with Cep57 and phenocopies the PCM disorganization by Cep57RNAi. These observations suggest that Calmodulin is not critically involved in mitotic PCM organization mediated by Cep57-PCNT interaction. However, given that we observed that addition of Calmodulin appeared to promote the Cep57-PCNT interaction at least *in vitro*, we cannot officially rule out the possibility that Calmodulin somehow facilitates the interaction for proper PCM arrangement in mitosis. We therefore mentioned and discussed this possibility in the revised manuscript (page 11 and 12, line 237-244).

Third, we apologize for not having properly described the way of doing this experiment. As this reviewer mentioned, the loading amount in lane1 for WB and coomassie blue staining was different, 1/30000 volume was loaded for WB, compared to the loading amount for coomassie blue staining. This was because of the fact that we would like to confirm the purity of recombinant HA-Cep57 in input lane of coomassie blue staining, and that we would like to make sure the molecular weight of HA-Cep57 in input lane of WB. However, the loading amount of coomassie blue staining would be too much in WB, so we reduce the loading amount of HA-Cep57 in WB.

5. Figure 3a: I cannot see the lagging chromosomes – resolution of the chromosomes is not good enough.

>We thank this reviewer for pointing this out. As this reviewer mentioned, the resolution of the data was not good enough, so we replaced it with the current representative data which more clearly shows the lagging chromosomes in Cep57-depleted cells (Figure 3a).

6. Do MVA patient cells also have lagging chromosomes?

>We sought to analyze the frequency of lagging chromosomes in MVA patient cells, but it was pretty difficult to gather enough number of mitotic MVA patient cells to judge a statistical significance. This is because the population of mitotic lymphoblast cells is generally small. In addition, we embedded

lymphoblast cells in smear gell, which decreased the resolution of DAPI staining and thus we could not observe lagging chromosomes very well, compared to usual staining condition. For these reasons, we did not show the quantification of the frequency of lagging chromosomes in MVA patient cells in this manuscript.

7. In Figure 4c there is no FLAG signal in the input.

>This reviewer is correct in pointing out that the signal of the input of FLAG-tagged Pericentrin is almost nothing. This was because of the technical problem that Pericentrin full-length is too large (~500 kDa) to efficiently detect by Westernblot. However, the point of this experiment is based on the ratio between immunoprecipitated fraction of Pericentrin and HA-Cep57 for testing a complex formation of Pericentrin and Cep57. This data explicitly indicates that the PINC domain is necessary for the Cep57-Pericentrin interaction (e.g. Cep57 FL/Pericentrin: Cep57 Δ 68-103/Pericentrin=352:1). We mentioned this technical difficulty in the corresponding figure legend for clarity (page 20, line428-430).

Comments on points raised by the reviewers

We thank all the reviewers of our original manuscript for their critical reading and for their useful and constructive comments (typed in blue), which we fully addressed in the revised version with new data (answers typed in black). Following their suggestions, we extensively performed all the experiments and substantially altered the manuscript, as detailed below.

Reviewers' comments:

Reviewer #1 (Remarks to the Author):

In this manuscript, Kitagawa and his colleagues reported roles of Cep57 in the centrosome. They revealed that Cep57 is located at the proximal end of mother centriole. Depletion of Cep57 causes PCM disorganization and precocious centriole disengagement during mitosis. As a result, the disengaged daughter centrioles function as spindle poles, leading to chromosome mis-segregation and aneuploidy. Similar defects were observed in mosaic variegated aneuploidy syndrome patient cells with cep57 mutations. They also observed a specific interaction of Cep57 with the PACT domain of pericentrin. Based on the results, the authors proposed that the Cep57-pericentrin complex provides a critical interphase between centriole core and PCM, which is crucial for the higher-order structure of the PCM.

This work revealed how Cep57 works at the centrosome. These results are novel and potentially important for elucidating structural and functional linkage between centrioles and PCM. Overall, their experiments seemed to be carefully carried out and the results looked trustful. This work is worth to be published in Nature Communication. The follows are major points to be resolved prior to publication.

> We thank this reviewer for the supportive comment on our manuscript.

1. The authors may have to explain why their results are different from the previous papers in which Cep57 functions in the kinetochore. One of critical differences between this work and Zhou et al. (2016) may be the immunostaining of Cep57 in mitotic cells. The Cep57 signals were coimmunostained with CREST, CENP-A and Mis12 in Zhou et al. (2016). On the other hand, this manuscript did not show an immunostaining image of the whole mitotic cells with the Cep57 antibody.

> This reviewer raised a critical question on the discrepancy of Cep57 localization at kinetochores. We followed the same protocol by which Zhou et al. detected Cep57 localization at kinetochores, except for using two different antibodies against Cep57. Although Cep57 signals were efficiently detected at centrosomes, we could not find any trace of Cep57 signal at kinetochores. We then tested different fixation methods by one of which we managed to detect Cep57 signals at kinetochores even though the signal intensity was weak (new data added in Figure S3d of the revised manuscript). We also performed fluorescence live cell imaging with Cep57-mCherry, but we hardly detected Cep57 signals at kinetochores (data not shown). We therefore conclude that Cep57 could modestly localize to kinetochores in human cells. However, contrary to the Cep57 knockdown phenotype that was reported in Zhou et al.

(2016), we could not observe significant defects in the spindle assembly checkpoint and mitotic duration (new data in Figure 3c and Supplementary Figure 3e,f). We therefore assume the specific function of Cep57 at kinetochores is likely very limited in the context of chromosome segregation. We are therefore sure that the PCM disorganization and precocious centriole disengagement provoked by Cep57 depletion should be irrelevant to potential function of Cep57 at kinetochores.

2. To enforce their proposal, knockdown-rescue experiments of Fig. 2f,g may be performed with HA-Cep57 deletion ($\Delta 68-103$) and truncated mutants (1-530; Zhou et al., 2016).

>We thank this reviewer for suggesting this experiment. Following this suggestion, we conducted the rescue experiment with HA-Cep57 deletion mutant lacking the PINC motif. As indicated in the original manuscript, the Cep57 mutant failed to localize to centrioles. Consistently, we demonstrated that expression of the Cep57 mutant did not rescue the phenotype of siCep57 (new data in Supplementary Figure 5k, mentioned in the revised manuscript (page 13, line 272-277)). Regarding the truncated mutant (1-530) that this reviewer mentioned, we suspect this might refer to the Mad1 mutant (Zhou et al. 2016) since Cep57 encompasses 500 a.a.

3. The authors reported that depletion of Cep57 did not lead to any defects in PCM formation and centriole duplication in interphase cells. It is worth to determine that depletion of Cep57 hardly affect daughter centriole recruitment of Cep152 in interphase cells.

>Prompted by this reviewer' comment, we tested the loading of Cep152 at centrioles upon Cep57 depletion, and found that it was not affected in interphase human cells (new data in Supplementary Figure 2a).

4. Precocious centriole disengagement was observed in Cep57-depleted cells (Figs. 3a). The separated centrioles with ectopic MTOC activities eventually became clustered to form pseudo-bipoles (Fig. 2h, Supplementary Fig. 2i,j). Nonetheless, the authors did not observe a delay in NEBD to anaphase onset time in Cep57-depleted cells (Fig. 3c). In fact, Zhou et al. (2016) reported a shortening of the NEBD to anaphase onset time in Cep57-depleted cells. The authors need to explain this observation, based on your proposal.

>This reviewer is correct in stating that mitotic delay could happen when the precociously separated centrioles acquire ectopic MTOC activities during mitosis. We carefully reanalyzed the live cell imaging movies with Cep57-depleted cells, and confirmed a significant mitotic delay when the cells exhibited chromosome misalignment because of ectopic MTOCs formation, including multipolar spindle formation (new quantification data in Supplementary Figure 3e). It seems that the precocious disengagement of centrioles itself was not sufficient to induce mitotic arrest, since these centrioles were clustered into pseudo-bipoles in a relatively quick manner. As this reviewer mentioned, Zhou et al. (2016) reported a shortening of the NEBD to anaphase onset time in Cep57-depleted cells. However, we could not observe such a significant shortening of mitosis in Cep57-depleted cells in our condition (Figure 3c). To address this further, we repeated the same protocol, as Zhou et al. (2016) performed, to measure the NEBD to anaphase

onset time under nocodazole-treatment, however, we could not observe a decrease in the mitotic duration, compared to the control, whereas the Mad2-depletion bypassed the SAC (spindle assembly checkpoint) pathway and so efficiently shortened the NEBD to anaphase onset time (new data in Supplementary Figure 3f). We therefore conclude that the function of Cep57 for the regulation of SAC seems to be limited and not that critical, at least, in our condition. This conclusion is more explicitly stated in the revised manuscript (page 8).

5. The model proposes that pericentrin interacts with Cep57 at the mother centriole. This configuration may be acceptable only in interphase cells. If the model is correct, then precocious centriole disengagement should be observed in interphase cells. According to Lawo et al. (2012), the PCM structure of the mitotic cells are quite different with massive accumulation of pericentrin and other components. Therefore, the model should be corrected for mitotic cells.

>We thank this reviewer for the valuable comment on our model and we understand the argument. We apologize for not having sufficiently explained the model, but we conclude that the Cep57-pericentrin interaction is crucial for the expansion and rearrangement of PCM and timely centriole disengagement during mitosis.

As prompted by this comment, we first re-examined the effect of Cep57 depletion on the loading of pericentrin in interphase and found that Cep57 depletion modestly affected the loading of pericentrin in interphase (new data in Fig. S5a,b). We therefore assumed that the loading of pericentrin is redundantly regulated by other factors in interphase. Indeed, we found the pericentrin Δ PACT mutant which lacks Cep57-binding region still localized to centrioles through its C-terminus (new data in Fig. S5c). We next searched for a new interactor responsible for the loading of pericentrin in interphase. Importantly, we found that Cep152 interacted with the C-terminus of pericentrin (but not with the PACT domain) and also that depletion of Cep152 significantly decreased the centriolar signal intensity of pericentrin in interphase (new data in Fig. S5d,e,f). These data indicate that Cep57 and Cep152, and perhaps other factors, redundantly regulate the loading of pericentrin in interphase. Unlike interphase, however, depletion of Cep57 causes precocious centriole disengagement and PCM disorganization during mitosis. This suggests that Cep57 is essential for the function of pericentrin during mitosis whereas the loading of pericentrin in interphase is redundantly regulated. The current model for PCM expansion is that additional PCM components are recruited on top of the layers of pre-existing interphase PCM. If the current model is true, precocious centriole disengagement and PCM disorganization would be observed already in interphase upon pericentrin or Cep57 depletion as this reviewer mentioned. However, we showed that this was not the case. Therefore, a new reasonable model should be presented to resolve this discrepancy. Our new model is that overall rearrangement of PCM components takes place for coupling PCM expansion and centriole engagement rather than just recruiting additional PCM components on top of the pre-existing PCM layers. In this process, our new data clearly indicate that a newly identified interaction between Cep57 and pericentrin is crucial for the mitotic rearrangement. It has been suggested that mitotic rearrangement of

PCM is dependent on Plk1, a major mitotic kinase. In line with this notion, we found that inhibition of Plk1 blocked precocious centriole disengagement during mitosis in Cep57-depleted cells (new data in Fig. S5l,m). Furthermore, we performed a new experiment to reinforce our model and found that a pericentrin chimera where the PACT domain was replaced with Cep57 (pericentrin-Cep57 chimera) efficiently rescued the disengagement phenotype of pericentrin depletion (new data in Fig. 5f,g,h). These data together provide the direct evidence that Cep57-pericentrin interaction is crucially important for the rearrangement of mitotic PCM. We accordingly added more discussion (page 14 and 15) and modified the model in Figure 6 so as to fit our claim.

6. The authors need to define how to discriminate precocious centriole disengagement.

>We thank this reviewer for raising this issue, which we defined in the revised manuscript (page 5, line 107-109). Basically, we discriminate precocious centriole disengagement in Cep57-depleted cells, when the distance between mother and daughter centrioles, in prometaphase or metaphase, is at least twice more than that of the control or interphase cells.

Reviewer #2 (Remarks to the Author):

The manuscript by Watanabe and colleagues explores the role of Cep57 in centriole engagement. They show that upon Cep57 depletion the mother-daughter centriole prematurely disengage as early as prophase, when centrioles normally disengage in late mitosis. The authors show that early disengagement leads to disorganized PCM, splayed spindle poles, and ultimately defects in chromosome segregation. Given that Pericentrin also shows a similar centriole disengagement and PCM phenotypes upon depletion, the authors explore the possibility that Cep57 and Pericentrin function together. They indeed show direct binding of these two proteins that depends on the newly identified and conserved PINC domain in Cep57 and the PACT domain in Pericentrin. The authors also suggest that the disease linked mutation K3154del in Pericentrin might cause the disease as a result of loss of interaction with Cep57.

This study is potentially interesting as it more clearly explains the previously shown multipolar spindle phenotype by identifying a centriole disengagement phenotype. As you will see below, the authors must do additional experiments to ensure that the two distant centrioles at the pole are, in fact, mother and daughter as opposed to two mother/daughter pairs. More importantly, the suggested finding in the paper is that Cep57 works directly with Pericentrin to ensure proper engagement until late mitosis. This model is not supported by the data. At this point, my 4 major concerns listed below preclude recommending this manuscript for publication.

>We appreciate this reviewer for supportive and constructive comments. As detailed below, we addressed his/her major and minor concerns in full.

Major concern

1. Measurements of the position of Cep57 and Cep192 should be repeated using SIM. Line-scans should be used to measure the outer diameter of each protein following a fit to a double gaussian curve. This will better represent the data and is necessary to compare to the common measurement found in centrosome literature. Additionally, detailed measurement methods should be described in the Methods section.

>We thank this reviewer for valuable suggestion on the detailed position of Cep57 within centrioles. Prompted by this comment, we performed STED microscopy analysis to precisely measure the outer diameter of each protein. The ring radius of Cep57 and Cep192 were measured using the ImageJ plugin "Radial Profile Extended". The obtained profile was then fitted with a Gaussian curve and the distance between the center of the ring and the peak of the Gaussian curve was defined as the radius. The diameter of the ring was defined as twice the radius. We again found that the diameter of Cep57 was close to that of Cep192. This method is more appropriate and is now explicitly described in the Methods section (page 38).

2. While it is likely that the two centrioles seen at the poles are the mother and daughter centrioles, the authors should stain mitotic cep57 mutant cells for a mother specific (appendage) protein to ensure these are not two mothers from a previous division error.

>As this reviewer mentioned, it would be important to make sure a pair of centrioles at the poles are indeed mother and its daughter centrioles. We therefore tested ODF2 staining which marks mother centrioles and found a single ODF2-positive focus out of two separate centrioles at a spindle pole in Cep57-depleted cells. This observation indicates that they are a pair of centrioles and supports the phenotype of precocious centriole disengagement. This new data is now added in Figure S2d and mentioned in the revised manuscript (page 5-6, line111-114).

3. I am not clear exactly how the authors dismiss an effect on SAC. If there is no change to the NEBD-anaphase timing and there are chromosome segregation errors, then it must be the case that loss of Cep57 suppresses SAC activation. Can the authors explain how this might occur? In the Mad1 experiment, was the level of Mad1 carefully measured under the Cep57 + nocodazole treatment? If a reduction is present, then SAC might not be properly activated. In addition, while the authors show average time of NEBD to anaphase is the same in control and knockdown cells, the distributions of values they obtain is different, with more cells with shorter times in the knockdown condition (Figure 3C). In Figure S3C, 2 of the 6 kinetochores shown do NOT have Mad1 on them, suggesting that the checkpoint is not active on these kinetochores. Finally, the authors say that Cep57 rarely localizes to kinetochores (line 148). The frequency at which this happens and the number of observations should be presented.

>This reviewer is correct in arguing that SAC activation and resulting mitotic delay should be induced when the separated centrioles with ectopic MTOC activity cause chromosome segregation error in Cep57-depleted cells. We carefully reanalyzed the live cell imaging movies of Cep57-depleted cells, and confirmed a significant mitotic delay when the cells exhibited apparent

chromosome segregation errors because of ectopic MTOCs formation, including chromosome misalignment, lagging chromosome and multipolar spindle formation (new quantification data in Figure S3e). This observation indicates that SAC is still active in the absence of Cep57. On the other hand, it seems that the precocious disengagement of centrioles itself was not sufficient to induce mitotic arrest, since these centrioles were clustered into pseudo-bipoles in a relatively quick manner. Therefore, as a whole, there was not significant increase in mitotic duration upon Cep57 depletion.

>Following the question of this reviewer, we quantified the Mad1 levels at kinetochores under nocodazole-treatment in Cep57-depleted cells. On the contrary to what Zhou et al. reported, the Mad1 signal intensity was not reduced and rather a bit increased, compared to the control (~1.1 fold, N>1000). So, as this reviewer pointed out, the original Figure S3c is not appropriate as a representative, we replaced it with the current panels in the revised manuscript (new data in Figure S3g,h).

>As this reviewer mentioned, Zhou et al. (2016) reported a shortening of the NEBD to anaphase onset time in Cep57-depleted cells. Following this reviewer's comment, we re-analyzed distribution of the values, however, we could not observe such a significant shortening of mitosis in Cep57-depleted cells in our condition (no statistical difference, Figure 3c). To address this further, we repeated the same protocol, as Zhou et al. (2016) performed, to measure the NEBD to anaphase onset time under nocodazole-treatment, however, we could not observe a decrease in the mitotic duration, compared to the control, whereas the Mad1-depletion bypassed the SAC pathway and so efficiently shortened the NEBD to anaphase onset time (new data in Figure S3f). We therefore conclude that the function of Cep57 for the regulation of SAC seems to be limited and not that critical, at least, in our condition. This conclusion is more explicitly stated in the revised manuscript (page 8).

> This reviewer raised a critical question on the discrepancy of Cep57 localization at kinetochores. We followed the same protocol by which Zhou et al. detected Cep57 localization at kinetochores, except for using two different antibodies against Cep57. Although Cep57 signals were efficiently detected at centrosomes, we could not find any trace of Cep57 signal at kinetochores. We then tested different fixation methods by one of which we managed to detect Cep57 signals at kinetochores (new data added in Figure S3d of the revised manuscript, this information is now explicitly described in page 8 and 27). We also performed fluorescence live cell imaging with Cep57-mCherry, but we hardly detected Cep57 signals at kinetochore (data not shown). We therefore conclude that Cep57 could modestly localize to kinetochores in human cells. However, contrary to the Cep57 knockdown phenotype that was reported in Zhou et al. (2016), we could not observe significant defects in the spindle assembly checkpoint during mitosis (new data in Figure 3c and S3f). We therefore assume the specific function of Cep57 at kinetochores is likely very limited in the context of chromosome segregation. We therefore think that the PCM disorganization and precocious centriole disengagement provoked by Cep57 depletion should be irrelevant to potential function of Cep57 at kinetochores.

4. The presented model of disengagement is not supported. The authors present a model where Cep57 recruits Pericentrin to the centriole to ensure

properly engaged centrioles. The data presented is not sufficient to support this mechanism. The authors would need to show the following at minimum to make this conclusion.

>We thank this reviewer for constructive suggestions on the following experiments.

- Show that loss of Cep57 leads to loss of PACT. This will establish that Cep57 is the *in vivo* anchor for Pericentrin through its PACT domain.

>We conducted this experiment and found that Cep57 depletion impaired centriolar loading of PACT, indicating that Cep57 is indeed an anchor for the PACT domain (new data in Figure 3h,l, mentioned in the revised manuscript (page 11, line 224-228)).

- Rescue the disengagement phenotype seen in Pericentrin loss with a Pericentrin chimera where the PACT domain is replaced with Cep57 within the context of full length Pericentrin.

>As this reviewer suggested this excellent experiment, we designed a construct of the Pericentrin chimera in which the PACT was replaced with Cep57. As expected, this mutant efficiently rescued the disengagement phenotype of Pericentrin knockdown, indicating that the Cep57-PACT interaction is sufficient for the function of Pericentrin and mitotic PCM organization. (new data in Figure 5f,g,h, and now mentioned in page 13, line 278-283).

- Show staining for Cep192 and Pericentrin in a Cep57 depleted cell and compare with RNAi to Pericentrin, Pericentrin delta-PACT, and K3154. Images from this manuscript do not indicate that Pericentrin localization is altered in Cep57 mutant as suggested. Depletion of cep57 causes cep192 to extend over 2um wide as shown in figure 2a. But the effect of cep57 on Pericentrin is much less as indicated by Figure 5f (maybe 300 nm change). This is also much different than the effect of K3154 and delta-PACT seen in figure 5d.

>Prompted by the comment from this reviewer, we reassessed the effect of Cep57 depletion on Pericentrin and Cep192 localization at centrosomes. We found that the patterns of Pericentrin and Cep192 at centrosomes were similarly disorganized and spread out upon Cep57 depletion (new data in Figure S5h). The extent of PCM dispersion was comparable to Pericentrin RNAi and expression of the Pericentrin mutants.

Other Concerns

1. The authors must better credit the previous work on Cep57 and its role in pole focusing. Although the two papers (Ref 19 and 20) are mentioned, the authors do not explicitly state that previous work has already shown an increase in multipolar spindles and unfocussed poles. They should state this explicitly in the introduction and again when discussing their own results, such as on line 123.

> We thank this reviewer for pointing this out. We now mention the previous finding on the role of Cep57 in pole focusing in the introduction of the revised manuscript (page 3). In addition, we mention their results more explicitly in the discussion (page 15).

2. The use of Crenolanib must be justified and specifically explained, as it is not a commonly used inhibitor. Additionally, details of this treatment must be

discussed in Methods section.

>We specified and explained the effect of Crenolanib on pole focusing in the revised manuscript (page 7, line 137-139). The treatment condition that we used in this study was also described in the method section and figure legends (page 36 and 26).

3. When discussing the measured percentage of multipolar spindles and chromosome segregation phenotypes, it would be important to address the discrepancy between the ~0.8% multipolar spindle phenotype and the 12.6% abnormal chromosome segregation phenotype. How does 0.8% lead to 12.6%? Are the authors suggesting that the tripolar spindles are transient, so called intermediates?

>We thank this reviewer for raising this issue. Regarding this, we have to apologize for having made a wording mistake in the sentence and for the confusion. The correct sentence in here is 'Live-cell imaging showed that Cep57-depleted cells exhibited chromosomal segregation errors (12.6%, compared to 2.6% in control cells) SUCH AS (not 'because of' in the original manuscript) chromosome misalignment, chromosome lagging and tripolar spindle formation, and also multi-nuclei formation' We now corrected the sentence in the revised manuscript (page 7, line 145-148) We assume that a pseudo-bipolar spindle with separate centriole pairs can induce chromosome missegregation. In other words, formation of multiple MTOCs could affect the fidelity of chromosome segregation even though they are clustered into two spindle poles. This could explain the difference between the ~0.8% multipolar spindle phenotype and the 12.6% abnormal chromosome segregation phenotype.

4. Line 193 and in corresponding legend – it is not clear what is meant by “target of the PACT domain”. Are the authors suggesting that Cep57 is the protein that anchors Pericentrin to the centriole via the PACT domain? If so, then this was not tested by the authors. To claim this, cells must be depleted of both Pericentrin and Cep57, while expressing a PACT-GFP construct. Loss of PACT at the centriole will strongly support Cep57 as the main anchor.

>As mentioned above, we conducted this experiment and found that Cep57 depletion abolished the centriolar recruitment of GFP-PACT (new data in Fig. 5). This observation supports the claim that Cep57 is the protein that mainly anchors Pericentrin to the centriole via the PACT domain. On the other hand, we observed that Pericentrin full-length still localized to centrioles even upon Cep57 depletion, suggesting that another anchor protein should load Pericentrin to centrioles through binding to other region (but not the PACT) of Pericentrin in interphase (new data in Fig. S5a,b,c). However, the Cep57-PACT of Pericentrin interaction is absolutely needed for proper arrangement of Pericentrin and PCM expansion during mitosis as the dispersion of Pericentrin in mitosis was observed upon Cep57 depletion (new data in Fig. S5h). We added the new data and mentioned this result in the revised manuscript (page 11, line 224-228 and page 13, line 270-272).

5. The authors should explicitly state that the K3154del might also disrupt other protein-protein interaction, especially that K3154 is present in one of the two Calmodulin binding domains of PACT.

>We appreciate the comment from this reviewer. It is known that two Calmodulin binding domains exist in the DPLP PACT domain in *Drosophila*, but in human, there seems to be only one Calmodulin binding domain in Pericentrin (c.f. Calmodulin Target Database). As far as we checked, K3154 does not reside within the Calmodulin binding domain of PACT of human Pericentrin (spanning 3193-3212 a.a.). However, we agree with this reviewer that K3154del might also disrupt other critical protein-protein interaction, so we modified this statement in the revised manuscript (page 12, lane 234-241).

6. Line 219 that authors state that PCM is disorganized, but figure S4a shows very organized PCM on each centriole.

>We apologize for the confusion and not having more clearly explained this phenotype. We observed PCM disorganization in the case of expression of the Pericentrin PACT mutants, but not Pericentrin knockdown solely (Figure 5d and S4a). Our interpretation is that Pericentrin depletion, but not expression of the PACT mutants, leads to loss of PCM expansion and this defect prevents the occurrence of PCM disorganization. It is also known that phosphorylation of pericentrin by Plk1 is needed for PCM expansion (Lee et al. 2011). This issue is now discussed in the revised manuscript (page 14 and 15). As this reviewer mentioned, in Figure S4a, PCM around centrioles is small, but organized upon Pericentrin depletion. In this situation, precocious centriole disengagement could happen, but not PCM disorganization because of loss of PCM expansion.

7. To properly measure the height of Pericentrin in figure 5f, the authors should include both a distal and proximal centriole marker, or a pan-centriolar marker, to ensure proper orientation and measurement accuracy. It might be the case the entire centriole is shorter. In the control, for example, Pericentrin looks like a ring, suggesting the authors were measuring Pericentrin diameter, not length.

>We appreciate this experimental suggestion. Accordingly, we tried triple-staining to mark distal and proximal ends of centrioles to ensure accurate orientation of centrioles. Unfortunately, we found that the far-red labeling of Cep152 antibodies to mark the proximal end gave us a fuzzy signal which was not appropriate for measuring the centriole length accurately. We then tested the staining with Cep152 and ODF2 antibodies to concurrently mark proximal and distal ends in the same color (red), however, it did not work as well. In the original experiment, we collected side views of centrioles by judging from the bar-shaped pattern of ODF2 (Tateishi et al. (2013) JCB, Fig. 4A; Ibi et al. (2011) JCS, Fig. 2B) and measured the height of Pericentrin in a vertical direction. Pericentrin should look like a ring only when we observe it from the top view and, in such case, ODF2 should also look like a ring. As far as we tested, this strategy worked the best for us to measure the Pericentrin height.

8. Mention of data from reference 28 should be removed as it is unpublished as far as I can tell.

> Thanks for the comment. Because we find that it was already published, we leave it as it is.

Minor

Line 94 – it is not clear what is meant by “at a constant rate”

>We replace “at a constant rate” with “occasionally”.

Line 96 - The authors should consider better discussing the huge evolutionary gap presented in Figure 1b. They show 5 vertebrates and one single celled eukaryote (a euglenoid). Is this protein present in any organisms between these? How might have Cep57 end up only in such evolutionarily separated species.

>We thank this reviewer for raising this important issue. As far as we looked into the existence of Cep57 among species, we found Cep57 homologues in most vertebrates, but apart from them, we could find it only in Trypanosoma and yeasts so far. Unfortunately, our blast search did not hit any other invertebrate organisms presumably due to the short consensus sequence. However, we speculate that its homologues may be present in other invertebrates. For instance, given that the PACT domains of DPLP (*Drosophila*) and Pericentrin (human) share homology, we speculate that a homologue of Cep57, the binding partner of Pericentrin, could exist in *Drosophila* as well. Accordingly, we now added more discussion, and mentioned that this protein is well conserved in vertebrates and also found in Trypanosoma and yeasts (page 4 and 15).

Line 136 – 139 – This is an assumption not supported by any data and should be removed or made clear that it is speculation.

>This reviewer is correct in stating that this claim seems an assumption. We now made it clear that this is an assumption by modifying these sentences. However, this claim is based on our observations as shown in Figure S2j and S3a. We indicated the figure number with this claim in the revised manuscript (page 7, lane 150-155).

Line 144 – given the two difference systems, I suggest replacing “contrary to the previous report” with “in contrast to the role of Cep57 in *Xenopus*” or something similar.

>The previous work which showed the role of kinetochore Cep57 in SAC activation was done in human cells (Zhou et al. (2016)), and we performed similar experiments using human cell lines. So we leave the phrase as it is.

Line 150 – remove “centriolar” from this sentence. Formally, the authors have shown that they can rescue by expressing Cep57, and this Cep57 can localize to the centriole, but they have not excluded the possibility that the rescue arises from its expression elsewhere in the cell.

>We thank this reviewer for the suggestion. We agree with this opinion and removed ‘centriolar’ from this sentence.

Line 178 – “implying” should be changed to “suggesting”

>We did, thank you very much.

Line 181-182 – while the authors show that PACT is necessary, they do not present data that it is sufficient for interaction with Cep57. Thus, the sentence

stating that 3139-3216 was capable of binding Cep57 should be changed to 3139-3336. The authors should also mention that they found that AA 1-1962 of Pericentrin, which does NOT contain PACT, also interacts with Cep57 (Figure 4d).

> We have changed the sentence to “short fragments of PCNT (aa 3139-3336 and aa 2945-3216) were capable of binding to Cep57” (page 10, line 201-204). Following the comment from this reviewer, we mentioned 1-1962 aa of Pericentrin also interacts with Cep57 in the revised manuscript (page 20, line 441-443).

Line 186-188 – similar to the above, the PACT domain per se was not tested for interaction with AKAP. Please change to the PACT-containing fragment of AKAP.

>Similarly, we changed it to the PACT-containing fragment of AKAP in the corresponding statement (page 10, line 206-209).

Line 226 – It would be best to remove the words “exact function” as the authors have not proven an exact mechanism for engagement or PCM organization.

>We followed this suggestion because we agree with the comment from this reviewer. Using “exact function” would not be appropriate here (page 13, line 284).

Reviewer #3 (Remarks to the Author):

The conserved Cep57-pericentrin module organizes PCM expansion and centriole engagement

Centrosomes are essential organelles that are duplicated once per cell cycle just like DNA. Centrosomes undergo a complex maturation process that ensures functioning of the daughter centrosome as microtubule organizing centre in the next cell cycle. Transition through mitosis is one of the key steps that is needed to decorate the daughter centriole from the previous S phase with pericentriolar material (PCM) such as pericentrin. Relatively little is now about this recruitment process.

Data on the function of CEP57 are heterogeneous and controversial. A 2007 publication in Cell suggests that CEP57 is a kinetochore component. Zhou et al published in Nature Communication that CE57 is a Mis12-interacting kinetochore protein involved in kinetochore targeting of Mad1-Mad2. He et al reported in JBC that CEP57 has a function in cytokinesis. Wu, Q. et al. reported that Cep57, a NEDD1-binding pericentriolar material component, is essential for spindle pole integrity. Interestingly, mutations in CEP57 cause mosaic variegated aneuploidy syndrome.

Watanabe et al analyse in this manuscript the function of CEP57. First, the authors identified a short 36 amino acid sequence that is conserved in the N-terminal coiled coil region of CEP57 family proteins. The authors next analyse CEP57 localizations at centrioles by STED microscopy and show correlation between CEP57 and pericentrin recruitment to the new mother centriole in G1. It seems that a region corresponding to the conserved part of CEP57 (PINC) is essential for centriole targeting. siRNA depletion of CEP57 caused PCM disorganization in mitosis with precociously-disengaged daughter centrioles but surprisingly the PCM of interphase cells was intact. This latter phenotype is inconsistent with the final model. Consistent with the precociously-disengaged daughter centrioles, HeLa cells with CEP57 depletion showed chromosomal segregation errors. They also address the question whether CEP57 has a role in the spindle assembly checkpoint (SAC). In contrast to a previous report in

Nature Communication, CEP57 depleted cells have the same mitotic duration as control cells. The authors then switch to MVA1 patient cells with biallelic reduction-of-function CEP57 mutations. MVA patient cells also showed precocious centriole disengagement in mitosis. In Figure 4 the authors test an interaction of CEP57 with pericentrin. Using overexpression conditions, they show co-immunoprecipitation of pericentrin and CEP57. Their data suggest that PINC region and PACT of pericentrin are important for this interaction. While the co-IP experiment suggests that CEP57 and pericentrin are present in common complexes, the *in vitro* binding data of Figure 4g are less convincing (see specific points). In the final Figure 5 they show that an amino acid deletion or partial deletion of the in PACT of pericentrin affects CEP57 interaction by IP. These mutations in pericentrin also premature centriole disengagement in mitosis as was seen upon CEP57 depletion.

This manuscript shows that the pericentrin-CEP57 interaction is important for PCM organization in mitosis. The PINC region of CEP57 and the PACT of pericentrin are important for PCM stabilization. However, the lack of interphase PCM organization argues against the simple CEP57-pericentrin recruitment model. Thus, presently it is unclear how CEP57 functions at centrosomes in mitosis. At the end, this manuscript does not substantially go beyond what has been published by He et al. in Cell Research. The conclusion of He et al. was: "Depletion of Cep57 leads to unaligned chromosomes and a multipolar spindle, which is induced by PCM fragmentation." Additional functional experiments on CEP57 and the regulation of CEP57 in mitosis are needed to justify publication in Nature Communication.

>We thank this reviewer for the constructive comments on our study. First, although our explanation might not be sufficiently clear in the original manuscript, our claim is that the Cep57-Pericentrin interaction is crucial for proper expansion and organization of mitotic PCM (Figures 2,3, and 5). Pericentrin without PACT domain can still localize to centrosomes, presumably by anchoring another centrosome protein except for Cep57, in interphase. It seems that PCM components are actively recruited to centrosomes and the binding mode of the components and its arrangement in the PCM change to fit proper expansion of functional PCM in mitosis. In this process, we found that the Cep57-Pericentrin interaction is critical because its

loss led to PCM disorganization and associated precocious centriole disengagement. A series of mutant analyses that we performed in Figure 4 and 5, clearly demonstrated that the specific interaction between Cep57 and Pericentrin PACT is needed for proper PCM expansion and timely centriole disengagement during mitosis. Moreover, the additional experiments further support the claim as detailed below.

> Moreover, prompted by this reviewer's comment, we re-examined the effect of Cep57 depletion on the loading of pericentrin in interphase and found that Cep57 depletion modestly affected the loading of pericentrin in interphase (new data in Fig. S5a,b). We therefore assumed that the loading of pericentrin is redundantly regulated by other factors in interphase. Indeed, we found the pericentrin Δ PACT mutant which lacks Cep57-binding region still localized to centrioles through its C-terminus (new data in Fig. S5c). We next searched for a new interactor responsible for the loading of pericentrin in interphase. Importantly, we found that Cep152 interacted with the C-terminus of pericentrin (but not with the PACT domain) and also that depletion of Cep152 significantly decreased the centriolar signal intensity of pericentrin in interphase (new data in Fig. S5d,e,f). These data indicate that Cep57 and Cep152, and perhaps other factors, redundantly regulate the loading of pericentrin in interphase. Unlike interphase, however, depletion of Cep57 causes precocious centriole disengagement and PCM disorganization during mitosis. This suggests that Cep57 is essential for the function of pericentrin during mitosis whereas the loading of pericentrin in interphase is redundantly regulated. The current model for PCM expansion is that additional PCM components are recruited on top of the layers of pre-existing interphase PCM. If the current model is true, precocious centriole disengagement and PCM disorganization would be observed already in interphase upon pericentrin or Cep57 depletion, as this reviewer pointed out. However, we showed that this was not the case. Therefore, a new reasonable model should be presented to resolve this discrepancy. Our new model is that overall rearrangement of PCM components takes place for coupling PCM expansion and centriole engagement rather than just recruiting additional PCM components on top of the pre-existing PCM layers. In this process, our new data clearly indicate that a newly identified interaction between Cep57 and pericentrin is crucial for the mitotic rearrangement. It has been suggested that mitotic rearrangement of PCM is dependent on Plk1, a major mitotic kinase. In line with this notion, we found that inhibition of Plk1 blocked precocious centriole disengagement during mitosis in Cep57-depleted cells (new data in Fig. S5l,m). Furthermore, we performed a new experiment to reinforce our model and found that a pericentrin chimera where the PACT domain was replaced with Cep57 (pericentrin-Cep57 chimera) efficiently rescued the disengagement phenotype of pericentrin depletion (new data in Fig. 5f,g,h). These data together provide the direct evidence that Cep57-pericentrin interaction is crucially important for the rearrangement of mitotic PCM. We accordingly added more discussion (page 14 and 15) and modified the model in Figure 6 so as to fit our claim.

>He et al. in Cell Research roughly described the phenotype of Cep57RNAi. Their observation is that, in Cep57-depleted cells, PCM fragmentation induces multipolar spindle formation. We uncovered a molecular mechanism that the Cep57-Pericentrin complex regulates mitotic PCM organization. Furthermore, although the conclusion of He et al. is that the PCM fragmentation is a cause

of multipolar spindle formation in Cep57-depleted cells, we conclude that the precocious centriole disengagement and resulting ectopic MTOC formation is the direct cause of the multipolar spindle formation. In fact, apparent multipolar phenotype is relatively rare, but a pseudo-bipolar spindle with multiple MTOCs is a major phenotype in Cep57-depleted cells, which can induce chromosome segregation error. Thus, our findings provide new insights into how PCM expansion and arrangement for timely centriole disengagement are coupled by revealing the function of the Cep57-Pericentrin complex in mitosis. Together with additional experiments that this reviewer requested, we believe that our results illustrate the new function of Cep57 at mitosis, as detailed below.

Major points.

1. Based on 36 aa homology in a coiled-coil region, I find it risky to conclude that CEP57 is a conserved factor across species.

>We thank the comment from this reviewer. As far as we looked into the existence of Cep57 among species, we found Cep57 homologues in most vertebrates (new data in Figure S1a), but apart from them, we could find it only in Trypanosoma and yeasts so far. Unfortunately, our blast search did not hit any other invertebrate organisms presumably due to the short consensus sequence. However, we speculate that its homologues may be present in other invertebrates. Accordingly, we now changed this statement in a modest manner, and mentioned that this protein is well conserved in vertebrates and also found in Trypanosoma and yeasts (page 4 and 15).

2. The authors conclude that CEP57 interacts with the PACT domain of PCNT. Therefore, CEP57 should co-localize with the C-terminus of PCNT but not the N-terminus of PCNT. To test this, STED analysis in Figure 1 needs to be performed with N- and C-terminal PCNT antibodies in comparison to CEP57. This should be done in interphase and mitosis. Mitosis is especially important because CEP57 depletion is causing mitotic phenotypes.

>We thank this reviewer for the valuable suggestion. Accordingly, we conducted STED analysis to test the co-localization of Cep57 and Pericentrin. As this reviewer suggested, using antibodies against N- and C-terminus of PCNT, we demonstrated that the position of C-terminus of PCNT is much closer to Cep57 at the inner centrosome than that of N-terminus of PCNT. This observation is true for both in interphase and mitosis. This new data is now added in Figure S4g and mentioned in the revised manuscript (page 11, line 228-233).

3. One of the main conclusions of this manuscript is that CEP57 recruits PCNT to the centriole. This model predicates that centriole binding of PCNT is defective in CEP57 depleted cells (interphase, perhaps mitosis). However, the authors state on lines 100-101 that “Cep57 did not lead to any defects in PCM formation and centriole duplication in interphase cells.” This finding is inconsistent with the model.

>We apologize for not having sufficiently explained this statement. Our model is that the Cep57-pericentrin interaction works for the mitotic expansion and rearrangement of PCM for proper centriole engagement until late mitosis. As already mentioned above, prompted by this reviewer’s comment, we first

re-examined the effect of Cep57 depletion on the loading of pericentrin in interphase and found that Cep57 depletion modestly affected the loading of pericentrin in interphase (new data in Fig. S5a,b). We therefore assumed that the loading of pericentrin is redundantly regulated by other factors in interphase. Indeed, we found the pericentrin Δ PACT mutant which lacks Cep57-binding region still localized to centrioles through its C-terminus (new data in Fig. S5c). We next searched for a new interactor responsible for the loading of pericentrin in interphase. Importantly, we found that Cep152 interacted with the C-terminus of pericentrin (but not with the PACT domain) and also that depletion of Cep152 significantly decreased the centriolar signal intensity of pericentrin in interphase (new data in Fig. S5d,e,f). These data indicate that Cep57 and Cep152, and perhaps other factors, redundantly regulate the loading of pericentrin in interphase. Unlike interphase, however, depletion of Cep57 causes precocious centriole disengagement and PCM disorganization during mitosis. This suggests that Cep57 is essential for the function of pericentrin during mitosis whereas the loading of pericentrin in interphase is redundantly regulated. The current model for PCM expansion is that additional PCM components are recruited on top of the layers of pre-existing interphase PCM. If the current model is true, precocious centriole disengagement and PCM disorganization would be observed already in interphase upon pericentrin or Cep57 depletion, as this reviewer pointed out. However, we showed that this was not the case. Therefore, a new reasonable model should be presented to resolve this discrepancy. Our new model is that overall rearrangement of PCM components takes place for coupling PCM expansion and centriole engagement rather than just recruiting additional PCM components on top of the pre-existing PCM layers. In this process, our new data clearly indicate that a newly identified interaction between Cep57 and pericentrin is crucial for the mitotic rearrangement. It has been suggested that mitotic rearrangement of PCM is dependent on Plk1, a major mitotic kinase. In line with this notion, we found that inhibition of Plk1 blocked precocious centriole disengagement during mitosis in Cep57-depleted cells (new data in Fig. S5l,m). Furthermore, we performed a new experiment to reinforce our model and found that a pericentrin chimera where the PACT domain was replaced with Cep57 (pericentrin-Cep57 chimera) efficiently rescued the disengagement phenotype of pericentrin depletion (new data in Fig. 5f,g,h). These data together provide the direct evidence that Cep57-pericentrin interaction is crucially important for the rearrangement of mitotic PCM. We accordingly added more discussion (page 14 and 15) and modified the model in Figure 6 so as to fit our claim.

4. In Figure 4 the authors show that CEP57 interacts with the PACT of PCNT. However these data have problems. First, binding efficiency is low (Fig. 4g). Second, binding is done with recombinant PACT without calmodulin binding. However, calmodulin is an essential co-factor of PACT. Third, HA-CEP57 in the Coomassie blue stained gel is a solid band. This amount should give a very strong signal in the immunoblot. However, in the immunoblot above the CEP57 signal in lane 1 is as in lane 5. CEP57 in lane 5 is not visible in the Coomassie Blue stained gel. Thus, the loading of the samples must be unequal.

>We thank this reviewer for the critical comments on the *in vitro* binding assay

with purified proteins of Cep57 and PCNT.

First, prompted by this reviewer's comment, we repeated this experiment in different conditions (e.g. buffer, protein amounts, loading volume, etc). The efficiency is now improved as shown in new Figure 4f. We also found that this interaction was reproducibly and robustly detected by different combination of tagging (GST or MBP) and yeast two-hybrid assay (new data in Figure 4f).

Second, we appreciate the comment on calmodulin as a cofactor of PACT. We tested the effect of Calmodulin on the PCM organization and centriole engagement in human cells, and did not find any significant defects upon Calmodulin depletion by two different siRNAs (new data in Figure S4h,i). Furthermore, although two Calmodulin binding domains exist in the DPLP PACT domain in *Drosophila*, there seems to be only one Calmodulin binding domain in human PCNT (c.f. Calmodulin Target Database). GFP-PACT Δ CaM that does not bind to Calmodulin was still able to localize to centrosomes in human cells (Gillingham et al. (2000) EMBO, Fig 3B). The PCNTK3154del mutant, the mutated residue of which resides outside of the Calmodulin binding motif (3193-3212 aa), is defective in interacting with Cep57 and phenocopies the PCM disorganization by Cep57RNAi. These observations suggest that Calmodulin is not critically involved in mitotic PCM organization mediated by Cep57-PCNT interaction. However, given that we observed that addition of Calmodulin appeared to promote the Cep57-PCNT interaction at least *in vitro*, we cannot officially rule out the possibility that Calmodulin somehow facilitates the interaction for proper PCM arrangement in mitosis. We therefore mentioned and discussed this possibility in the revised manuscript (page 11, line 234-241).

Third, we apologize for not having properly described the way of doing this experiment. As this reviewer mentioned, the loading amount in lane1 for WB and coomassie blue staining was different, 1/30000 volume was loaded for WB, compared to the loading amount for coomassie blue staining. This was because of the fact that we would like to confirm the purity of recombinant HA-Cep57 in input lane of coomassie blue staining, and that we would like to make sure the molecular weight of HA-Cep57 in input lane of WB. However, the loading amount of coomassie blue staining would be too much in WB, so we reduce the loading amount of HA-Cep57 in WB.

5. Figure 3a: I cannot see the lagging chromosomes – resolution of the chromosomes is not good enough.

>We thank this reviewer for pointing this out. As this reviewer mentioned, the resolution of the data was not good enough, so we replaced it with the current representative data which more clearly shows the lagging chromosomes in Cep57-depleted cells (Figure 3a).

6. Do MVA patient cells also have lagging chromosomes?

>We sought to analyze the frequency of lagging chromosomes in MVA patient cells, but it was pretty difficult to gather enough number of mitotic MVA patient cells to judge a statistical significance. This is because the population of mitotic lymphoblast cells is generally small. In addition, we embedded lymphoblast cells in smear gel, which decreased the resolution of DAPI staining and thus we could not observe lagging chromosomes very well,

compared to usual staining condition. For these reasons, we did not show the quantification of the frequency of lagging chromosomes in MVA patient cells in this manuscript.

7. In Figure 4c there is no FLAG signal in the input.

>This reviewer is correct in pointing out that the signal of the input of FLAG-tagged Pericentrin is almost nothing. This was because of the technical problem that Pericentrin full-length is too large (~500 kDa) to efficiently detect by Westernblot. However, the point of this experiment is based on the ratio between immunoprecipitated fraction of Pericentrin and HA-Cep57 for testing a complex formation of Pericentrin and Cep57. This data explicitly indicates that the PINC domain is necessary for the Cep57-Pericentrin interaction (e.g. Cep57 FL/Pericentrin: Cep57 Δ 68-103/Pericentrin=352:1). We mentioned this technical difficulty in the corresponding figure legend for clarity (page 20, line436-438).

Reviewers' comments:

Reviewer #2 (Remarks to the Author):

I have now read the updated manuscript and the response to reviewers. I must commend the authors for the additional experiments they performed, especially the STED imaging and chimera experiment. Overall, the authors have addressed my concerns and I therefore support publishing this work.

Reviewer #3 (Remarks to the Author):

The conserved Cep57-pericentrin module organizes PCM expansion and centriole engagement

This revised manuscript by Watanabe et al. addresses many points that I have raised during the first reviewing process. It demonstrates that the protein Cep57 has a function in centriole engagement, directly or indirectly via pericentrin.

Having said this, the manuscript still has some problems. I do not agree with the title. The title is misleading. The authors have not shown that Cep57 is a conserved protein. It is a centrosomal protein with a short conserved motif. Whether the "Cep57-pericentrin module" is conserved has not been investigated in this manuscript.

Another aspect that is still open is the Cep57-PACT interaction. The authors propose a direct interaction. However, at the end all experiments are either based on the overexpression of CEP57 and PACT/pericentrin or on in vitro binding experiments with recombinant proteins. The overexpression experiments show that Cep57 and PACT are probably present in common complexes. The in vitro binding experiments show a weak interaction in the range of less than 1:30.000 of the input. Is this specific? Some other control than GST/MBP would have been important. The analysis of the localization of PCNT C and Cep57 in Fig. S4g does not suggest co-localization of both proteins. Sure, both form a ring close to centrioles. However, the intensity distribution along this ring is not the same. A peak of Cep57 intensity is accompanied by a minimum in PCNT C intensity. This is not what I expect if Cep57 and PACT directly interact.

In Fig. S5 the authors show that PCNT intensity at the mother centrosome is not strongly affected by Cep57 depletion. They explain this finding by a second binding site for PCNT and suggest that Cep152 provides this site. However, the data in Fig. S5d and e, especially the picture Fig. 5e and the IP Fig. S5d, sgCep152, are not convincing. Depletion of Cep152, although it impacts PCNT stronger than Cep57 depletion (according to Fig. S5b) does not have an influence on centriole engagement (see Hatch et al JCB 2010). Also Drosophila asterless mutants only mildly affect PCM function, which does not fit to Fig. S5f. A rescue experiment would be important to show that sgCep152 specifically targets the Cep152 locus.

Comments on points raised by the reviewers

We thank all the reviewers for their critical reading and for their useful and constructive comments (typed in blue). In particular, following the comments by the reviewer #3, we performed additional new experiments and fully addressed all the concerns in the re-revised version with new data (answers typed in black). Thus, we altered the manuscript, as detailed below.

Reviewers' comments:

Reviewer #2 (Remarks to the Author):

I have now read the updated manuscript and the response to reviewers. I must commend the authors for the additional experiments they performed, especially the STED imaging and chimera experiment. Overall, the authors have addressed my concerns and I therefore support publishing this work.

> We thank this reviewer for the supportive comment on our revised manuscript.

Reviewer #3 (Remarks to the Author):

The conserved Cep57-pericentrin module organizes PCM expansion and centriole engagement

This revised manuscript by Watanabe et al. addresses many points that I have raised during the first reviewing process. It demonstrates that the protein Cep57 has a function in centriole engagement, directly or indirectly via pericentrin.

> We thank this reviewer for the critical comment on our revised manuscript.

Having said this, the manuscript still has some problems. I do not agree with the title. The title is misleading. The authors have not shown that Cep57 is a conserved protein. It is a centrosomal protein with a short conserved motif. Whether the "Cep57-pericentrin module" is conserved has not been investigated in this manuscript.

> We agree with this suggestion and changed the title as follows; "The

Cep57-pericentrin module organizes PCM expansion and centriole engagement.”

Another aspect that is still open is the Cep57-PACT interaction. The authors propose a direct interaction. However, at the end all experiments are either based on the overexpression of CEP57 and PACT/pericentrin or on *in vitro* binding experiments with recombinant proteins. The overexpression experiments show that Cep57 and PACT are probably present in common complexes. The *in vitro* binding experiments show a weak interaction in the range of less than 1:30.000 of the input. Is this specific? Some other control than GST/MBP would have been important. The analysis of the localization of PCNT C and Cep57 in Fig. S4g does not suggest co-localization of both proteins. Sure, both form a ring close to centrioles. However, the intensity distribution along this ring is not the same. A peak of Cep57 intensity is accompanied by a minimum in PCNT C intensity. This is not what I expect if Cep57 and PACT directly interact.

> Prompted by this comment, we repeated the *in vitro* binding experiments by using the C-terminus PCNT lacking the PACT domain as a new specific negative control other than GST/MBP. The new data clearly showed that Cep57 interacts with the C-terminus of PCNT, but not with the C-terminus of PCNT lacking the PACT domain. This suggests that PCNT directly interacts with Cep57 via the PACT domain. We now added the new data and mentioned this result in the re-revised manuscript (new data in Fig. S4d, line 213-215).

> As this reviewer mentioned, the intensity distribution along the ring of Cep57 or PCNT in Fig. 4g seems to be uneven. However, there is a clear tendency that the signal of Cep57 overlapped with that of the PCNT C-terminus, but not with that of the PCNT N-terminus. This observation is true for both interphase and mitotic cells although the PCNT signal in mitosis is a bit fuzzy perhaps due to PCM expansion. Accordingly, we now changed this statement in a modest manner (line 233-235).

In Fig. S5 the authors show that PCNT intensity at the mother centrosome is not strongly affected by Cep57 depletion. They explain this finding by a second binding site for PCNT and suggest that Cep152 provides this site. However, the data in Fig. S5d and e, especially the picture Fig. 5e and the IP Fig. S5d, sgCep152, are not convincing. Depletion of Cep152, although it impacts PCNT

stronger than Cep57 depletion (according to Fig. S5b) does not have an influence on centriole engagement (see Hatch et al JCB 2010). Also *Drosophila* asterless mutants only mildly affect PCM function, which does not fit to Fig. S5f. A rescue experiment would be important to show that sgCep152 specifically targets the Cep152 locus.

> We thank this reviewer for pointing this out. We checked the Cep152 paper (Hatch et al. 2010) that this reviewer mentioned, but we could not find the data supporting that Cep152 depletion does not have an influence on centriole engagement. Unfortunately, it is not possible to test if Cep152 affects centriole engagement because Cep152 is required for centriole duplication as well. Regarding *Drosophila* Cep152 (Asterless), some paper mentioned the role of Cep152/Asterless on PCM function (Fu et al. NCB 2016; Novak et al. Dev Cell 2016), so we believe that Cep152 and Asterless have a similar role in PCM function.

> As this reviewer suggested, we performed a rescue experiment with CRISPR-resistant GFP-Cep152 and found that overexpression of the GFP-Cep152 protein efficiently rescued the phenotype (reduction of PCNT signal) provoked by sgCep152. This suggests that sgCep152 specifically targets the Cep152 locus and also that the resulting phenotype reflects the function of Cep152 on PCM organization. The new data is added in the re-revised manuscript (new data in Fig. S5e-f, line 256-258).